# Learning Division with Neural Arithmetic Logic Modules

## Abstract

To achieve systematic generalisation, it first makes sense to master simple tasks such as arithmetic. Of the four fundamental arithmetic operations $(+,-,\times,\div)$, division is considered the most difficult for both humans and computers. In this paper we show that robustly learning division in a systematic manner remains a challenge even at the simplest level of dividing two numbers. We propose two novel approaches for division which we call the Neural Reciprocal Unit (NRU) and the Neural Multiplicative Reciprocal Unit (NMRU), and present improvements for an existing division module, the Real Neural Power Unit (Real NPU). Experiments in learning division with input redundancy on 225 different training sets, find that our proposed modifications to the Real NPU obtains an average success of $85.3\%$ improving over the original by $15.1\%$. In light of the suggestion above, our NMRU approach can further improve the success to $91.6\%$.

## 1   Introduction

Imagine you must learn to divide 2 numbers, but are only given 10 numbers and the target value. This task requires finding the 2 relevant operands, the order to divide the operands, and learning to divide. In machine learning, this is equivalent to a supervised regression task where the aim is to learn the underlying function between the inputs and output such that the solution is generalisable to any input. The ability to select relevant features is a desirable property of neural networks, useful for improved intepretability, reduced pre-processing costs and greater generalisation [Chandrashekar and Sahin, 2014]. The ability to model division, one of the four fundamental arithmetic operations, is necessary for expressing dynamical systems [Sahoo et al., 2018], and physics-based formulas [Udrescu and Tegmark, 2020]. However, even recent models still struggle to learn division when there is input redundancy [Schlör et al., 2020].

The main challenge of the above task comes from learning the selection and operation at the same time, which can lead to conflicting priorities when learning network weights. Furthermore, the natural properties of division of values around zero leads to undesirable gradients. Models which deal with this naively (e.g. MLPs) are unable to deal with the fluctuant gradients caused by the asymptotic nature and discontinuities in division [Trask et al., 2018].

Can we build models which can learn division in the presence of its undesirable, yet valid, properties? We aim to address this question in this paper. Specifically, we contribute the following:[1]

- **Improvements to the Real NPU** [Heim et al., 2020] including: clipping, discretisation and constrained initialisation to improve performance in learning division on different training ranges.

---

[1]Code (MIT license) available at: `https://anonymous.4open.science/r/nalu-stable-exp-neurips-review-2E4C/`.

Submitted to 35th Conference on Neural Information Processing Systems (NeurIPS 2021). Do not distribute.

- **Two novel division modules**, the NRU and the NMRU. The NRU explores extending the NMU weight ranges from [0,1] to [-1,1] to include division, where we find a weakness in learning from negative ranges. Learning from the weaknesses of the NRU, the NMRU extends the NMU to learn division while keeping weights values between [0,1]. We further boost performance by using a Real NPU inspired sign retrieval mechanism, enabling the NMRU to gain the best performance when using a mean squared error (MSE) loss.

- **New understanding into the hindrances in learning division** including: training on mixed-sign inputs, training on negative ranges, and division on extremely small values. We find these difficulties can be sufficiently identified using synthetic division tasks.

The broader impact of our work relates to interpretable Artificial Intelligence where our modules can be included in larger networks for applications such as image classification or analogy creation, whilst retaining the ability to produce transparent generalisable solutions. However, there are possible negative societal impacts. Such modules can be viewed as specialised feature selectors/aggregators which do not require integrating domain knowledge. Therefore, if a non-domain-expert tries interpreting relations in the input data, they may incorrectly interpret causality, which can be especially harmful if such a case occurs on medical or financial data. Mitigating against such downstream issues requires to first focus efforts on producing robust modules to different distributions and understand their affect on learning other networks architectures (e.g. CNN). Understanding this will enable recognising situations where these modules can aid and where they should avoid being used.

## 2   Related Work

One approach to learn division would be symbolic regression networks [Sahoo et al., 2018]. However, a symbolic approach pre-defines the operations, which is not a limitation of using Neural Arithmetic Logic Modules (NALMs).

NALMs are neural networks which learn arithmetic operations and input selection [Mistry et al., 2021]. The weights of these networks are intepretable such that a discrete value represents a specific operation. For example, '-1' to represent division and '0' for no selection. From this research field, we focus on the Real NPU and the NMU. Until now, the Real NPU only has learned division on training ranges of either $\mathcal{U}[0.1,2]$ or Sobol(0,0.5) [Heim et al., 2020]. It remains unclear if this module is robust to other training ranges even as a stand-alone unit. Robustness to training ranges is important as these module's applicational use comes from being part of larger end-to-end networks, where the input range into the module cannot be controlled. The NMU is a multiplication module which we extend to also do division. The authors of the NMU believe such an extension incurs too many limitations for learning [Madsen and Johansen, 2020]. We use this paper as an opportunity to explore this belief.

Trask et al. [2018] developed the Neural Arithmetic Logic Unit (NALU) which can model all four arithmetic operations. However, studies show this module to be unstable in learning division [Schlör et al., 2020, Heim et al., 2020]. In particular, their gating method responsible for selecting an operation cannot learn consistently [Madsen and Johansen, 2020]. Schlör et al. [2020] developed iNALU additionally applying weight and gradient clipping, sign retrieval, regularisation, reinitialisation and separating shared parameters to the NALU. Even with these modifications, they still find consistently learning division to a high precision to remain unattainable. Furthermore, Heim et al. [2020]'s results imply iNALU is outperformed by the Real NPU for division.

## 3   Architectures

This section introduces the architectures for the (Real) NPU, NRU, and the NMRU. The (Real) NPU is an existing module, which we improve in Section 5. The NRU and NMRU are novel contributions. Appendix A summarises the important properties of these division modules.

### 3.1   Real Neural Power Unit

Heim et al. [2020] develop a module to learn to multiply and divide, using the intuition from Trask et al. [2018] that *multiplicative operations are additive operations in log space*. Their work extends this idea into complex space. The NPU can be used with its complex form (Equation 1) requiring both

a complex and real weight matrix ($\boldsymbol{W}^{(i)}, \boldsymbol{W}^{(r)}$), or only its real form the Real NPU (Equation 2). For improved gradients, a relevance gate $\boldsymbol{r}$ (Equation 3) is used which converts inputs close to 0 (i.e. irrelevant features) to 1 to avoid the resulting output evaluating to 0. A gating vector $\boldsymbol{g}$, learns to select relevant input elements, where gate values are clipped between [0,1] during training.

$$\text{NPU} := \exp(\boldsymbol{W}^{(r)} \log(\boldsymbol{r}) - \boldsymbol{W}^{(i)} \boldsymbol{k}) \odot \cos(\boldsymbol{W}^{(i)} \log(\boldsymbol{r}) + \boldsymbol{W}^{(r)} \boldsymbol{k}), \tag{1}$$

$$\text{RealNPU} := \exp(\boldsymbol{W}^{(r)} \log(\boldsymbol{r})) \odot \cos(\boldsymbol{W}^{(r)} \boldsymbol{k}) \tag{2}$$

$$\text{where} \quad \boldsymbol{r} = \boldsymbol{g} \odot (|\boldsymbol{x}| + \epsilon) + (\boldsymbol{1} - \boldsymbol{g}) \quad \text{and} \quad k_i = \begin{cases} 0 & x_i \geq 0 \\ \pi g_i & x_i < 0 \end{cases}. \tag{3}$$

A weighted L1 penalty is used when training. The weight value $\beta$ grows between predefined values $\beta_{start}$ to $\beta_{end}$ and is increased every $\beta_{step} = 10,000$ iterations by a growth factor $\beta_{growth} = 10$. We focus on the Real NPU over the NPU as the solution of the tasks in this paper can be captured using only real values meaning that the complex form is not required.

### 3.2 Neural Reciprocal Unit

We propose the NRU, which can model multiplication and division. We extend the NMU, motivated by *division being multiplication of reciprocals*. The range which weight values can be is extended from [0,1] to [-1,1], where -1 represents applying the reciprocal on the corresponding input element. A NRU output element $z_o$ is defined as

$$\text{NRU} : z_o = \prod_{i=1}^{I} \left( \text{sign}(x_i) \cdot |x_i|^{W_{i,o}} \cdot |W_{i,o}| + 1 - |W_{i,o}| \right), \tag{4}$$

where $I$ is the number of inputs. Assuming weights are either 1 (multiply) or -1 (reciprocal), $|x_i|^{W_{i,o}}$ will apply the operation on an input element. The absolute value is used so that the module only operates in the space of real numbers, as $x_i^{W_{i,o}}$ for a negative input ($x_i$) when $-1 < W_{i,o} < 1$ results in a complex number. The use of absolute means the sign of the input must be reapplied. For the no-selection case $W_{i,o} = 0$, we want the input element to convert to 1 (the identity value), resulting in applying $\cdot |W_{i,o}| + 1 - |W_{i,o}|$. The derivative of the absolute function at 0 is undefined meaning the gradients of Equation 4 can contain points of discontinuity. To alleviate this issue, we approximate the absolute function using a scaled $\tanh$ (inspired by Faber and Wattenhofer [2020]). More formally,

$$|W_{i,o}| = \begin{cases} \tanh(1000 \cdot W_{i,o})^2 & \text{if training} \\ |W_{i,o}| & \text{otherwise} \end{cases}.$$

The scale factor (1000) controls how close to the absolute function the approximation is, where larger values give a more accurate approximation. For clipping and regularisation, the same scheme as the Neural Addition Unit (NAU) (see Appendix B) is used.

### 3.3 Neural Multiplicative Reciprocal Unit

An alternate extension of the NMU, also motivated by *division being multiplication of reciprocals* is the NMRU (Equation 5). We concatenate the reciprocal of the input (plus a small $\epsilon$) to the input resulting in a module which only needs to learn selection. Hence, weights can be in the range [0,1].

$$\text{NMRU} : z_o = \prod_{i=1}^{2I} (W_{i,o} \cdot |x_i| + 1 - W_{i,o}) \cdot \sum_{i=1}^{2I} (\cos(W_{i,o} \cdot k_i)) \text{ , where } k_i = \begin{cases} 0 & x_i \geq 0 \\ \pi & x_i < 0 \end{cases}. \tag{5}$$

The iteration over $2I$ represents the going through all inputs and their reciprocals. We calculate the magnitude and sign separately, joining the result at the end. The magnitude is calculated passing absolute of the concatenated input through an NMU architecture and the sign by using a cosine mechanism similar to the Real NPU. However, unlike the Real NPU only the weight matrix is required. The norm of the weight's gradients are clipped to 1 prior to being updated by the optimiser. This is done to alleviate the issue of exploding gradients caused by including the reciprocal to the inputs. For clipping and regularisation, the same scheme as the NMU (see Appendix B) is used.

Table 1: Interpolation (train/validation) and extrapolation (test) ranges used. Data (as floats) is drawn from a Uniform distribution with the range values as the lower and upper bounds.

| | | | | | |
|---|---|---|---|---|---|
| **Interpolation** | [-20, -10) | [-2, -1) | [-1.2, -1.1) | [-0.2, -0.1) | [-2, 2) |
| **Extrapolation** | [-40, -20) | [-6, -2) | [-6.1, -1.2) | [-2, -0.2) | [[-6, -2), [2, 6)] |
| **Interpolation** | [0.1, 0.2) | [1, 2) | [1.1, 1.2) | [10, 20) | |
| **Extrapolation** | [0.2, 2) | [2, 6) | [1.2, 6) | [20, 40) | |

## 4  Experiment Setup

We introduce the two main experiments used to evaluate modules, including: default parameters, train and test ranges, and evaluation metrics. The tasks evaluate the ability of a single module to divide two numbers from an input vector in two settings: **no redundancy** and **with redundancy**.

**Default parameters:**   All experiments use a mean squared error (MSE) loss with an Adam optimiser [Kingma and Ba, 2015], with 10,000 samples for the validation and test sets. The best model for evaluation is taken using early stopping on the validation set. All runs are over 25 different seeds. All inputs are required in the *no redundancy* setting, i.e., input size of 2. Training takes 50,000 iterations where each iteration consists of a different batch of size 128. The Real NPU uses a learning rate of 5e-3 with sparsity regularisation scaling during iterations 40,000 to 50,000. The NRU and NMRU use sparsity regularisation scaling during iterations 20,000 to 35,000 and a learning rate of 1 and 1e-2 respectively. In contrast, the *redundancy* setting uses an input size of 10, where 8 input values are not required for the final output. The total training iterations are extended to 100,000 with batch sizes of 128. The learning rates for the Real NPU, NRU and NMRU are 5e-3, 1e-3 and 1e-2 respectively. Sparsity regularisation scaling occurs during iteration 50,000 to 75,000 for all modules. A summary of all relevant parameters is found in Appendix C.

**Ranges:**   The interpolation (train/validation) and extrapolation (test) ranges, are found in Table 1. The chosen ranges are influenced by Madsen and Johansen [2020].

**Evaluation metrics:**   We use the Madsen and Johansen [2019]'s evaluation scheme, consisting of three evaluation metrics: the success on the extrapolation dataset against a near optimal solution (*success rate*), the first iteration which the task is considered solved (*speed of convergence*), and the extent of discretisation towards the weights' inductive biases (*sparsity error*). Sparsity error calculated by $\max_{i,o}(\min(|W_{i,o}|, 1 - |W_{i,o}|))$, measures the weight element which is the furthest away from the acceptable discrete weights for the module. A success means the MSE of the trained model is lower than a threshold value (i.e. the MSE of a near optimal solution). We differ from Madsen and Johansen [2019] by using a fixed threshold value 1e-5 rather than a simulated MSE, as there are no intermediate layers to accumulate numerical errors. We choose this precision as it can be guaranteed when working with 32-bit PyTorch Tensors. 95% confidence intervals (over the 25 seeds) are calculated from a specific family of distributions dependant on the metric. The success rate uses Binomial distribution because trials (i.e. run on a single seed) are either pass/ fail situations. The convergence metric uses a Gamma distribution and sparsity error uses a Beta distribution. Both Beta and Gamma can easily approximate the normal distribution and support its corresponding metric.

## 5  Improving the Real NPU's Robustness

We first improve the robustness of the Real NPU on different training ranges. We use the Single Module Task with no redundancy (see Section 4) to investigate the following questions:

1. Is L1 regularisation required, and if so, do the regularisation parameters require tuning?
2. Does clipping the weight matrix aid learning?
3. Does enforcing discretisation on parameters improve convergence?
4. Can the weight matrix initialisation be improved?

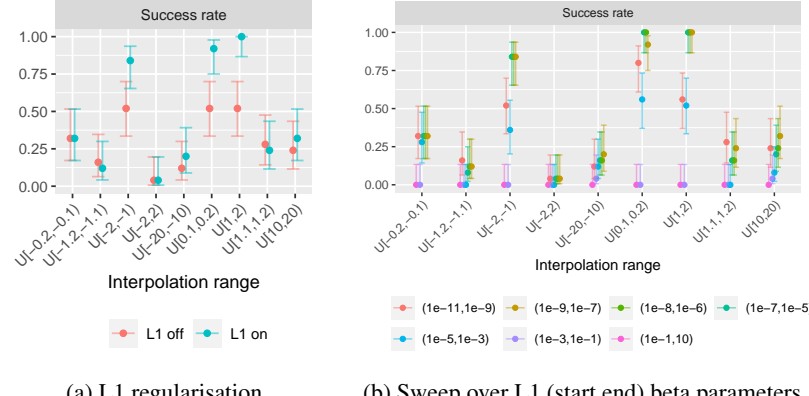

(a) L1 regularisation

(b) Sweep over L1 (start,end) beta parameters

Figure 1: Exploring the effect and sensitivity of L1 regularisation on the Real NPU

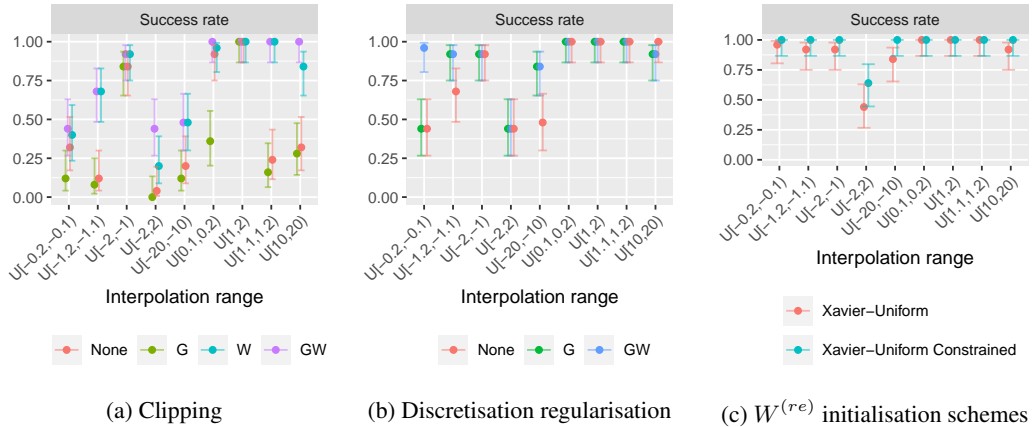

(a) Clipping

(b) Discretisation regularisation

(c) $W^{(re)}$ initialisation schemes

Figure 2: Effect of clipping, discretisation, and the NAU initialisation scheme on the Real NPU.

To address each question in order, we propose applying incremental modifications to the Real NPU. These modifications include: ablation study on the L1 regularisation (including a sweep over the scaling range hyperparameters), clipping, enforcing discretisation, and a more restrictive initialisation scheme. We assume that we are optimising the Real NPU to perform multiplication or division. Therefore, *we trade-off the flexibility of having non-discretised weights, which enables the success of modelling the SIR data in Heim et al. [2020, Section 4.1] , in favour of sparse models with discrete weight values.* All the modifications suggested can also be generalised for the NPU architecture.

**Is L1 regularisation required? (Yes)** L1 encourages sparsity (i.e., zero weights) in solutions. Zero-valued weights means not to select an input and return the identity value 1. For the task, the optimal weight values require selecting all inputs and therefore non-zero values, suggesting the application of L1 could be damaging. Therefore, we compare against a model which does not use L1 regularisation, shown in Figure 1a. Removing L1 proves to be detrimental in five of the nine cases shown and only shows minor improvements in two of the nine ranges (i.e., $\mathcal{U}$[-1.2,-1.1] and $\mathcal{U}$[1.1,1.2]). Hence, we keep L1 regularisation. The L1 regularisation scaling (see Section 3.1), requires setting the hyperparameters for the start ($\beta_{start}$) and end ($\beta_{end}$) scaling values. We run a sweep over six different start and end values, denoted (<start>, <end>), displaying results in Figure 1b. We find the configuration (1e-9, 1e-7) is the most successful when considering performance on all the ranges, and larger scaling values perform worse.

**Does clipping the learnable parameters help? (Yes)** Division and multiplication operations are represented by weight values of -1 and 1 respectively. The current architecture does not constrain the weights which can result in large weight values. The gate weights do get clipped and saved to another

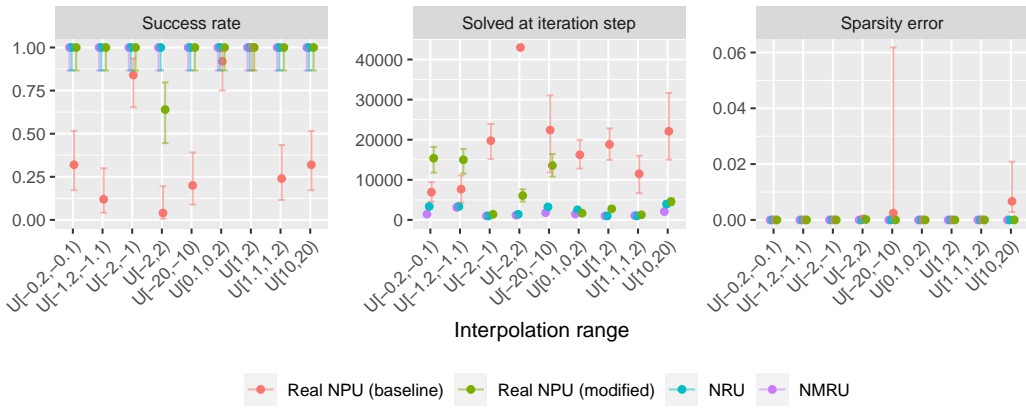

Figure 3: Division without redundancy (input size 2).

variable during the forward pass, meaning after an update step the gate values can also be out of the range [-1,1]. Hence, we investigate the effect of applying clipping directly to the weight and gate values after every optimisation step. Results, shown in Figure 2a, show clipping is beneficial, with clipping on both weight and gate (or just on the weights) to improve over the baseline on all ranges (excluding $\mathcal{U}[1,2)$ where the baseline has already achieved full success).

**Does enforcing discretisation help? (Yes)** Modelling division in a generalisable manner requires all learnable parameters to be discrete i.e., a value from {-1, 0, 1}. Using Madsen and Johansen [2020]'s regularisation scaling scheme, we penalise weights for not being discrete. We modify the scaling factor to be $\hat{\lambda} = 1$ and the regularisation to go from 'off' to 'on' between iterations 40,000 to 50,000. Results, shown in Figure 2b, show discretising the gate improves over the baseline but also discretising the weights is additionally beneficial (especially for range $\mathcal{U}[-0.2,-0.1)$). $\mathcal{U}[10,20)$ is the only range where the baseline outperforms using discretisation, succeeding on two additional seeds.

**Does using a more constrained initialisation help? (Yes)** $W^{(r)}$ uses a Xavier-Uniform initialisation [Glorot and Bengio, 2010]. This can result in weights initialised out of the range [-1,1]. Therefore, we use the initialisation for the Neural Addition Unit which is a constrained form of the Xavier-Uniform that does not allow the fan values of the uniform distribution to go beyond 0.5, meaning that no weight value will be out of the range [-1,1] [Madsen and Johansen, 2020]. Figure 2c shows using the constrained initialisation provides improvements over multiple ranges.

## 6 Results: Single Module Task

We analyse the results for the: Real NPU without using the modifications of Section 5, Real NPU with modifications, NRU, and NMRU.

### 6.1 No Redundancy

Figure 3 shows the baseline Real NPU without modifications struggles with all ranges except $\mathcal{U}[1,2)$, struggling with sparsity on the larger ranges. Applying the modifications deals with the sparsity issue and improves the robustness such that only range $\mathcal{U}[-2,2)$ struggles (with a success rate of 0.64). The NRU and NMRU achieve full success over all ranges while solving the problem consistently fast and with low sparsity error. The success of the NRU is correlated with the learning rate (see Appendix E).

### 6.1.1 Mixed-signed Inputs

The remaining failure range of the Real NPU is $\mathcal{U}[-2,2)$ where inputs can consist of arbitrary signed values (e.g. all positives, all negatives, or a mixture of positive and negative values). *We question if the failure is due to the input samples in a batch having different signs from each other, or if the problem is due to the fact data samples can be close to 0 (leading to singularity issues).* To investigate

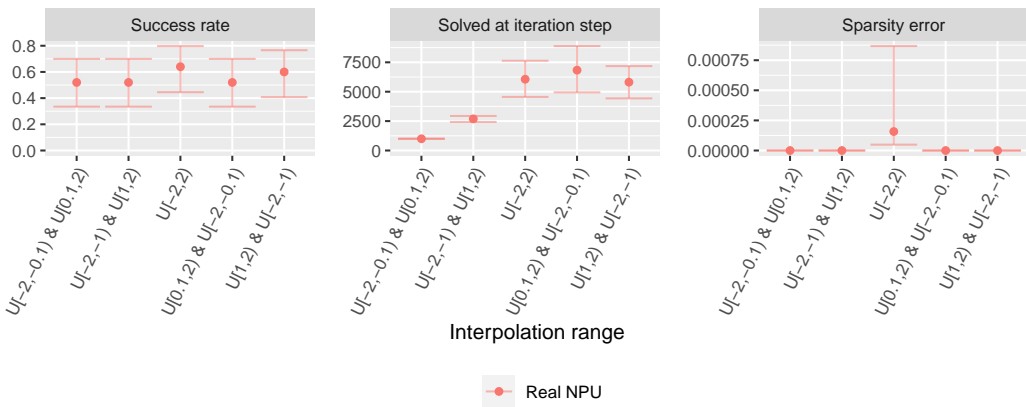

Figure 4: Extrapolation results on training the Real NPU using mixed-sign datasets that control the sign of the input elements. The ranges are in order of the datasets (i.e. dataset 1 to 5).

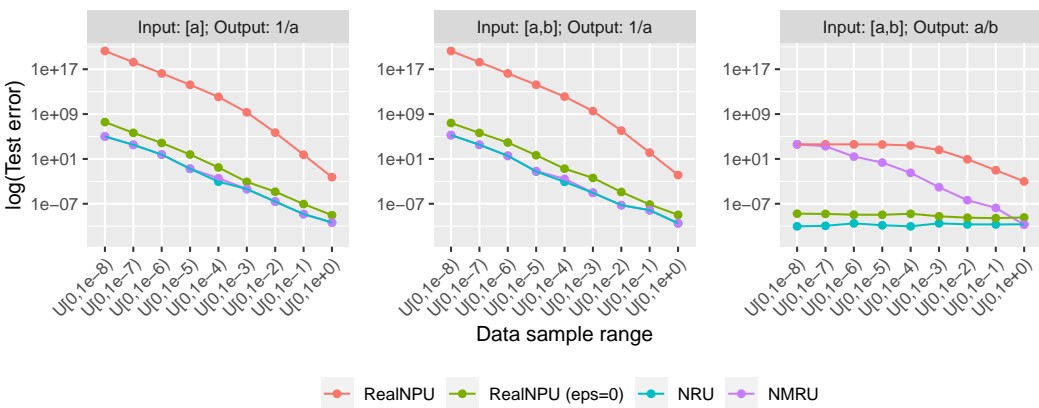

Figure 5: Effect of the singularity issue on the Real NPU, NRU and NMRU over increasing input ranges. Left: Reciprocal for an input size of 1 (no redundancy). Middle: Reciprocal for an input size of 2 (with redundancy). Right: Division for an input size of 2 (no redundancy).

this, we create additional mixed-sign datasets, controlling the range for each element in the input. The interpolation and extrapolation ranges for the different datasets can be found in Appendix C. Datasets 1, 2, 4 and 5 sample a positive value for one input element and a negative value for the other element. Dataset 3 samples the signs randomly. Datasets 2 and 5 avoid sampling close to 0 values to mitigate the singularity issue. As shown by Figure 4, the Real NPU struggles on all these ranges, implying that the core issue is not from different input samples having different signs or due to the input samples being able to contain small values close to 0. The underlying issue is therefore most likely correlated to the each element in an input having different signs. When the denominator of the output is positive (dataset 1 or 2), the solution is found faster than when the denominator is a negative value (dataset 4 or 5). When the signs for an input element are controlled, discretisation/sparsity is no problem, in contrast when the signs are arbitrary the sparsity error are slightly (though not significantly) higher.

## 6.2 Division by Small Numbers

Division by zero remains a challenge to model due to the inability to provide an computational value for the output and gradient. Furthermore, the discontinuous nature at zero causes its neighbouring values to have large gradients. To understand the extent of this issue when learning, we explore learning to divide by values close to zero using three tasks with increasing difficulty: 1) learning to take the reciprocal of a single input, 2) taking the reciprocal of the first input given two inputs, and 3) diving the first input by the second given two inputs. Figure 5 plots the test error for different modules

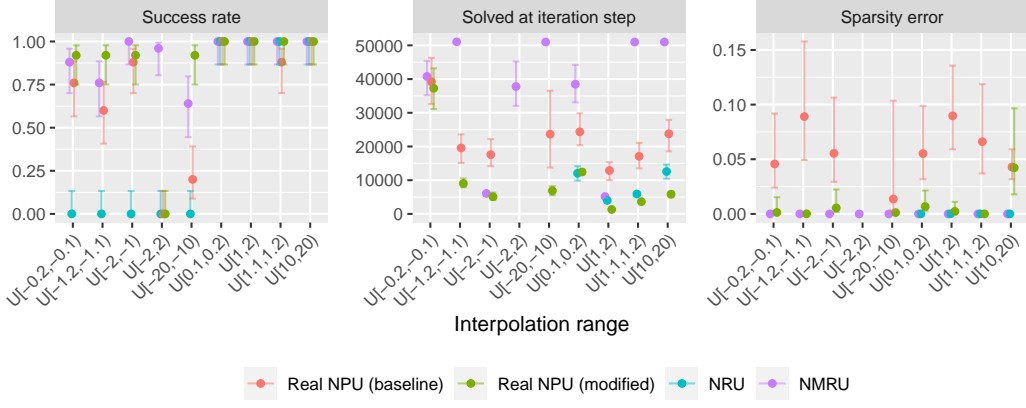

Figure 6: Division with redundancy (input size 10).

assuming the module weights are set to the 'gold' solution for the three tasks. As the range values become closer to zero, the test error thresholds become increasingly large. Therefore, even with the correct weights, relying on the test errors alone as an indicator become increasingly deceptive with values close to zero. The Real NPU has larger test errors for all tasks and ranges, caused by adding $\epsilon$ to the input (see Equation 3). Setting $\epsilon = 0$ reduces the test error at the cost of the ability to deal with zero-valued inputs. Appendix F provides the corresponding experimental results for these tasks.

## 6.3 With Redundancy

Introducing redundancy (Figure 6) causes failure modes to arise. Failures on range $\mathcal{U}$[-2,2) become more prevalent. The baseline Real NPU produces high sparsity errors relative to the other modules suggesting struggle with discretisation. Using the modified Real NPU improves over all ranges of the baseline (which were not already at full success) in terms of success, speed and sparsity.[2] To ensure that complex weights do not fix the issue, we test the NPU module with all the modifications used on the real weight matrix (see Appendix G). Complex weights hinders success and convergence speeds of negative ranges. Assuming the global solution only uses the real weights, we enforce the complex weights to be clipped between [-1,1] and to go to 0 during the regularisation stage using a L1 penalty. This did not result in any significant improvements against the Real NPU results. Input redundancy effects the NRU the most, resulting in full failures on all the negative ranges. The NMRU is the only module with success for the range $\mathcal{U}$[-2,2), which is a result of using the sign mechanism (see Appendix H). It performs well over all ranges though can be outperformed by the modified Real NPU for negative ranges. Multiple ranges for the NMRU are solved around 50,000 iterations correlating to the sparsity regularisation being turned on.

### 6.3.1 Gradient Difficulties with the NRU

The partial derivative for the NRU weights, Equation 6, can give insight to the struggles of the NRU.

$$\frac{\partial \hat{\mathbf{y}}}{\partial w_i} = \tanh(1000w_i)(\text{sign}(x_i)|x_i|(\tanh(1000w_i)\log(|x|)+ \\ 2000\,\text{sech}(1000w_i)^2) - 2000\,\text{sech}(1000w_i)^2) \times \text{NRU}_{\tilde{\mathbf{x}}\in\mathbf{x}\setminus\{\mathbf{x_i}\}}(\tilde{\mathbf{x}}). \tag{6}$$

$\text{NRU}_{\tilde{\mathbf{x}}\in\mathbf{x}\setminus\{\mathbf{x_i}\}}(\tilde{\mathbf{x}})$ applies the NRU to all inputs excluding $x_i$ influencing the gradient values between subsequent update steps. Factoring out this term, the following observations are made. If $x_i \approx 0$ and $w_i \approx 0$ then gradients become increasingly large. If $x_i \approx 0$ and $-1 \leq w_i < 0$ then as $w_i \to -1$ all gradients for $x_i$ where $|x_i| >> 1$ become increasingly small. The gradients for $x_i = -1$ and $x_i = 1$ are 0 regardless the value of $w_i$. If $w_i = 0$ then the gradient is 0 for all $x_i$, a result of using the $\tanh$ approximation. Even if the sign and magnitude are calculated separately and then combined (see Appendix I) to try to control the gradient better, the problem remains. Therefore, we conclude that extending the NMU to divide using a weight of -1 is a poor choice when there are redundant inputs.

---

[2]Except for the sparsity error for range $\mathcal{U}$[10,20).

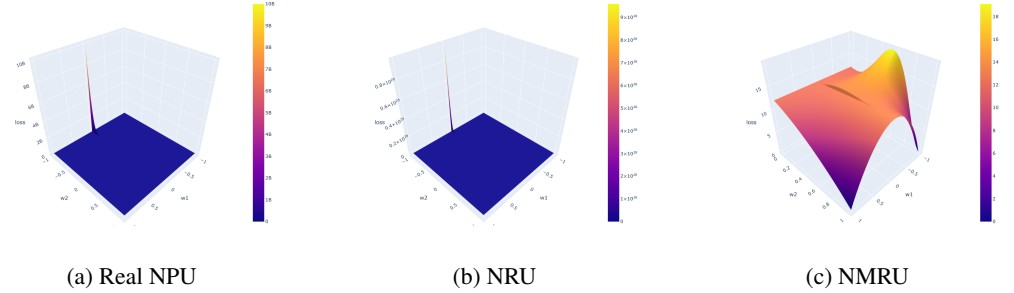

|  (a) Real NPU | (b) NRU | (c) NMRU |

Figure 7: Root Mean Squared loss curvature for the NAU stacked with either a RealNPU, NRU, or NMRU. "The weight matrices are constrained to $\mathbf{W}_1 = \begin{bmatrix} w_1 & w_1 & 0 & 0 \\ w_1 & w_1 & w_1 & w_1 \end{bmatrix}$, $\mathbf{W}_2 = \begin{bmatrix} w_2 & w_2 \end{bmatrix}$. The problem is $(x_1 + x_2) \cdot (x_1 + x_2 + x_3 + x_4)$ for $x = (1, 1.2, 1.8, 2)$" [Madsen and Johansen, 2020]. The ideal solution is $w_1 = w_2 = 1$, though other valid solutions do exist e.g., $w_1 = -1, w_2 = 1$. (The NMRU's weight matrix would be $\mathbf{W}_2 = \begin{bmatrix} w_2 & w_2 & 0 & 0 \end{bmatrix}$, and the Real NPU's $\mathbf{g} = \begin{bmatrix} 1 & 1 \end{bmatrix}$. )

### 6.3.2 The Real NPU's and NMRU's Exploitation of Multiplicative Rules

The NMRU solutions exploit the inverse rule of division in that $a_i \cdot \frac{1}{a_i} = 1$. Since the input also contains the reciprocals, numerous extrapolative solutions exist. However this comes at the cost of finding a 'simple' solution which contains ones only for relevant inputs. The Real NPU exploits the rules $a_i \cdot 0 = 0$ and $1^{a_i} = 1$ enabling non-zero weight values if the corresponding gate value is 0. However, we can avoid this by allowing 0 to also not be penalised during sparsity regularisation stage (see Appendix G). We find this alleviates the exploitation issue with no cost to performance.

## 7 Discussion

In this paper, we demonstrate the limitations of intepretable neural networks in learning to divide. Using the no redundancy setting (size 2), we find that the Real NPU is challenged when training data consists of mixed-signed inputs even with our applied improvements. Increasing the difficulty to have an input redundancy (with 8 redundant and 2 relevant input values) magnifies this issue, but also introduces failure modes for the NRU and NMRU for negative ranges. The NRU is unable to handle any negative ranges, in which we conclude it is not wise to use with MSE. Alternate losses can improve certain failure cases though sometimes at the cost of performance on other ranges. For further details see Appendix J which displays results on a correlation and scale-invariant based loss.

Our NMRU is the only module with reasonable success over all tested ranges, requiring only $2I \times O$ learnable parameters. However, this comes at the cost of the simplicity of the solution due to its exploitation of the identity rule; an issue the Real NPU does not have.

Once robust modules are attainable in a single layer setting, the next step would be to question performance when learning stacked modules, e.g. learning a stacked additive and multiplicative module. Previously, Madsen and Johansen [2020, Figure 2] illustrates the troubles for multiplicative models with the capacity for division. They show how a stacked summative-multiplicative module can lead to an exploding loss when the output of the summative module is close to 0 and the multiplicative model tries to divide. In Figure 7, we recreate their setup to produce the loss surfaces for the NAU-Real NPU[3], NAU-NRU and NAU-NMRU respectively.[4] We find a similar issue with the Real-NPU and NRU, as both these units use a weight range of [-1,1]. In contrast, the NMRU, whose weight's range is limited to [0,1] does not have exploding losses.

In conclusion, division remains a challenge to learn using intepretable neural networks, even for the simplest tasks. Nevertheless, by identifying the specific areas causing difficulty (e.g., training ranges), and useful architecture properties (e.g., using a sign retrieval mechanism), we hope the community has better intuition for dealing with division and develop more robust modules to learn division.

---

[3]The NAU is a summative module [Madsen and Johansen, 2020].

[4]Appendix K displays larger versions of these plots.

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
