# A Properties of a Division Module

When building a division module, the following properties should be included:

**Ability to multiply:** Without multiplication the module is limited to expressing reciprocals.

**Intepretable weights:** Having a discrete set of weight values to represent specific operations, e.g., -1 to divide, 1 to multiply, 0 to not select. Doing this also has the additional benefit of producing generalisable solutions to out-of-bounds data.

**Calculating the output:** This can be decomposed into three tasks: input selection, magnitude calculation, and sign calculation.

**Magnitude:** This is achieved using discrete weights. The Real NPU and NRU use -1 for reciprocals, 1 for multiplication. The NMRU uses 1 for selecting an input element which represents either a multiplication or reciprocal depending on its position.

**Sign of the output:** Calculating the sign value (1/-1) of the output can occur at an element level in which the sign is calculated for each intermediary value as each input element is being processed, or at the higher input level in which the sign is calculated separately for the magnitude and then applied once the final output magnitude is calculated. The NRU uses the prior method while the Real NPU and NMRU use the latter method. If an input is 0 or considered irrelevant then the output sign will be 1. (Ablation studies on the NMRU, Figure 7, suggest the latter option which separately calculates the sign to be more beneficial).

The Real NPU and NMRU use the cosine function to calculate the final sign of the module's output neuron. Below shows the state diagram of how the sign value (i.e. the state) of the output would change depending on the inputs and relevant parameters being processed. We only consider the discrete parameters for simplicity. Both the Real NPU and NMRU use the same state diagram but have different conditions for a state transition to occur.

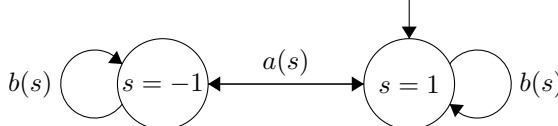

The conditions for the Real NPU transition functions $a(s) = -s$ and $b(s) = s$, where $s$ is the state value -1, or 1, are defined as follows:

$$a(s) : x_i < 0 \land w_{i,o} \in \{-1, 1\} \land g_i = 1 \; ,$$
$$b(s) : x_i \geq 0 \lor w_{i,o} = 0 \lor g_i = 0 \; .$$

Transitioning from one sign to another only occurs if the input element ($x_i$) is negative and is considered relevant i.e. the gate ($g_i$) and weight value ($w_{i,o}$) is non-0. In contrast, to remain at a state requires either the input element to be $\geq 0$ or not be considered relevant.

The conditions for the NMRU transition functions $a(s) = -s$ and $b(s) = s$, where $s$ is the state value -1, or 1, are defined as follows:

$$a(s) : x_i < 0 \land w_{i,o} = 1 \; ,$$
$$b(s) : x_i \geq 0 \lor w_{i,o} = 0 \; .$$

Transitioning from one sign to another only occurs if the input element ($x_i$) is negative and is considered relevant i.e. the weight value ($w_{i,o}$) is 1. To remain at a state requires either the input element to be $\geq 0$ or the weight value to not select the input.

**Selection:** Not all inputs are relevant for the output value. To process any irrelevant input elements can be interpreted as converting to the identity value of multiplication/division (=1). The identity property means that any value multiplied/divided by the identity value remains at the original number. Hence, irrelevant inputs are converted into 1 (rather than being masked out to 0). For the multiplication case, this stops the output becoming 0, and for division it avoids the divide by 0 case. For all the explored modules, a weight value of 0 will deal with the irrelevant input case. However, the Real NPU goes a step further by also having an additional gate vector with the purpose of learning to select relevant inputs. Such gating has been proven to be helpful for an NPU based module [Heim et al., 2020], but may not be necessary when dealing with weights between [0,1] like in the NRMU (see Appendix H).

## B  Neural Addition and Neural Multiplication Units' (NAU & NMU)

Madsen and Johansen [2020] develop two modules: one for dealing with addition and subtraction (the NAU) and the other for multiplication (the NMU). NAU output element $a_o$ is defined as

$$\text{NAU} : a_o = \sum_{i=1}^{I} (W_{i,o} \cdot \mathrm{x}_i) \tag{1}$$

where $I$ is the number of inputs. The NMU output element $m_o$ is defined as

$$\text{NMU} : m_o = \prod_{i=1}^{I} (W_{i,o} \cdot \mathrm{x}_i + 1 - W_{i,o}). \tag{2}$$

Before passing a input through a module, the weight matrix is clamped to [-1,1] for the NAU or [0,1] for the NMU. Weights are ideally discrete values, where the NAU is 0, 1, or -1, representing no selection, addition and subtraction, and the NMU is 0 or 1, representing no selection and multiplication. To enforce discretisation of weights both units have a regularisation penalty for a given period of training. The penalty is

$$\lambda \cdot \frac{1}{I \cdot O} \sum_{o=1}^{O} \sum_{i=1}^{I} \min \left( |W_{i,o}|, 1 - |W_{i,o}| \right), \tag{3}$$

where $O$ is the number of outputs and $\lambda$ is defined as

$$\lambda = \hat{\lambda} \cdot \max \left( \min \left( \frac{iteration_i - \lambda_{start}}{\lambda_{end} - \lambda_{start}}, 1 \right), 0 \right). \tag{4}$$

Regularisation strength is scaled by a predefined $\hat{\lambda}$. The regularisation will grow from 0 to $\hat{\lambda}$ between iterations $\lambda_{start}$ and $\lambda_{end}$, after which it plateaus and remains at $\hat{\lambda}$.

## C  Experiment Parameters

Tables 1 and 2 for the breakdown of parameters used in the Single Module Tasks. Table 3 gives the interpolation and extrapolation ranges used in the mixed-sign datasets tasks.

Table 1: Parameters which are applied to all modules. Parameters have been split based on the experiment. $^*$Validation and test datasets generate one batch of samples at the start which gets used for evaluation for all iterations. $^\dagger$ the Real NPU modules use a value of 1.

| Parameter | Without redundancy | With redundancy |
|---|---|---|
| **Layers** | 1 | 1 |
| **Input size** | 2 | 10 |
| **Total iterations** | 50,000 | 100,000 |
| **Train samples** | 128 per batch | 128 per batch |
| **Validation samples**$^*$ | 10000 | 10000 |
| **Test samples**$^*$ | 10000 | 10000 |
| **Seeds** | 25 | 25 |
| **Optimiser** | Adam (with default parameters) | Adam (with default parameters) |
| $\hat{\lambda}^\dagger$ | 10 | 10 |

Table 2: Parameters specific to the Real NPU modules for the Single Module Tasks.

| Parameter | Value |
|---|---|
| $(\beta_{start}, \beta_{end})$ | (1e-9,1e-7) |
| $\beta_{growth}$ | 10 |
| $\beta_{step}$ | 10000 |
| $\hat{\lambda}$ | 1 |

Table 3: Mixed-Sign Datasets: The interpolation and extrapolation ranges to sample the two input elements for a single data sample. The target expression to learn is: input 1 $\div$ input 2.

| | INTERPOLATION | | EXTRAPOLATION | |
|---|---|---|---|---|
| **DATASET** | **INPUT 1** | **INPUT 2** | **INPUT 1** | **INPUT 2** |
| 1 | U[-2, -0.1) | U[0.1, 2) | U[-6, -2) | U[2, 6) |
| 2 | U[-2, -1) | U[1, 2) | U[-6, -2) | U[2, 6) |
| 3 | U[-2, 2) | U[-2, 2) | U[-6, -2) | U[2, 6) |
| 4 | U[0.1, 2) | U[-2, -0.1) | U[2, 6) | U[-6, -2) |
| 5 | U[1, -2) | U[-2, -1) | U[2, 6) | U[-6, -2) |

### C.1  Parameter Initialisation

We give the initialisations used on the different module parameters:

**Real NPU**: The real weight matrix uses the Pytorch's Xavier Uniform initialisation. The gate vector initialises all values to 0.5. (This is the same initialisation used in Heim et al. [2020].)

**NPU**: The imaginary weight matrix is initialised to 0. The rest of the parameters are initialised same as the Real NPU. (This is the same initialisation used in Heim et al. [2020].)

**NRU**: The weight matrix uses a Xavier Uniform initialisation which can have a maximum range between -0.5 to 0.5 (depending on the network sizes). (This is the same initialisation the Neural Addition Unit uses [Madsen and Johansen, 2020].)

**NMRU**: The weight matrix uses a Uniform initialisation which can have a maximum range between 0.25 to 0.75 (depending on the network sizes). (This is the same initialisation the Neural Multiplication unit uses [Madsen and Johansen, 2020].)

# D  Hardware and Time to Run Experiments

All experiments were trained on the CPU, as training on GPUs takes considerably longer. All Real NPU experiments were run on XXX (the University of XXX 's supercomputer), where a compute node has 40 CPUs with 192 GB of DDR4 memory which uses dual 2.0 GHz Intel Skylake processors. All NRU and NMRU experiments were run on a 16 core CPU server with 125 GB memory 1.2 GHz processors.

Table 4 displays time taken for each experiment to run a single seed for a single range. Timings are based on a single run rather than the runtime of a script execution because the queuing time from jobs when executing scripts is not relevant to the experiment timings. For a single model, a single experiment would have 225 runs (for 9 training ranges and 25 seeds).

Table 4: Timings of experiments.

| Experiment | Model | Approximate time for completing 1 seed (mm:ss) |
|---|---|---|
| No redundancy (size 2) | Real NPU | 03:20 |
| | NRU | 02:00 |
| | NMRU | 03:00 |
| With redundancy (size 10) | Real NPU | 05:30 |
| | NRU | 05:00 |
| | NMRU | 05:15 |

## E    NRU on the Single Module Task (no redundancy): Effect of Learning Rate

Figure 1 displays the effect of different learning rates for the NRU. An learning rate of 1 gets full success on all ranges with performance deteriorating as the learning rate reduces.

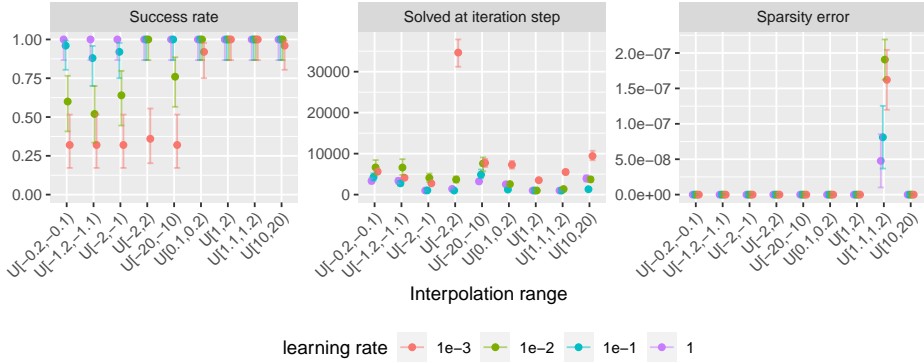

Figure 1: Different learning rates on the NRU for the Single Module Task (no redundancy)

## F   Division by small values: Experimental Results

This section shows the results on trying to learn the reciprocal/division of values close to zero using the Real NPU, NRU and NMRU. We train and test on the ranges where the lowest bound is 0 and the upper bounds are: 1e-4, 1e-3, 1e-2, 1e-1 and 1. Unless stated otherwise, the hyperparameters of a model are set to what is used for the Single Layer Task without redundancy. The first task runs for 5,000 iterations with no regularisation for any module. The second and third tasks both run for 50,000 iterations.

Due to precision errors, a solution with the ideal parameters will not evaluate to a MSE of 0. Therefore, we calculate thresholds which the test MSE should be within. A threshold value for a task is calculated from evaluating the MSE of each range's test dataset for each module, using the 'golden' weight values and adding an epsilon term[1] to the resulting error which takes into account precision errors. All experiments are run using 32-bit precision.

In general, successful runs take longer to solve as the input ranges become smaller. The simplest task, of taking the reciprocal when the input size is 1 (Figure 2) is achieved with ease for all modules, though for $\mathcal{U}[0,1e\text{-}4]$, we find the NRU begins to start struggling.

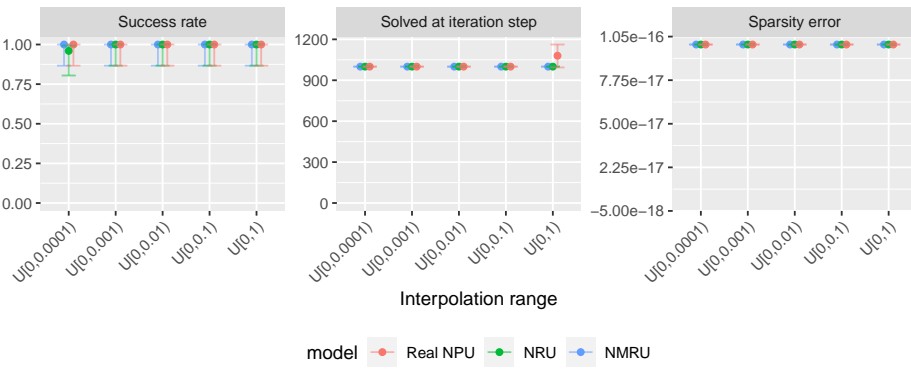

Figure 2: Input: [a], output $\frac{1}{a}$. Learns reciprocal when there is no input redundancy.

Introducing a redundant input (Figure 3) greatly impacts performance with only the NMRU able to achieve reasonable success for the larger ranges. The successes shown for the Real NPU at range $\mathcal{U}[0, 1e\text{-}4]$ are false positives caused by the $\epsilon$ in the architecture used for stability. Test false positives can also be indicated by the high sparsity error of the weights.

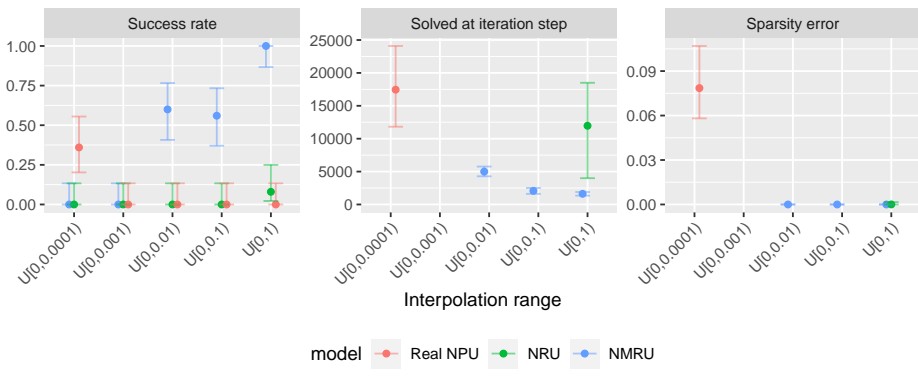

Figure 3: Input: [a,b], output $\frac{1}{a}$. Learns reciprocal of the first input when there is redundancy.

Modifying the task to division (Figure 4), meaning the redundant input is now relevant, shows improvement for the NMRU and NRU for the larger ranges.

---

[1]The term is the pytorch default eps value, torch.finfo().eps

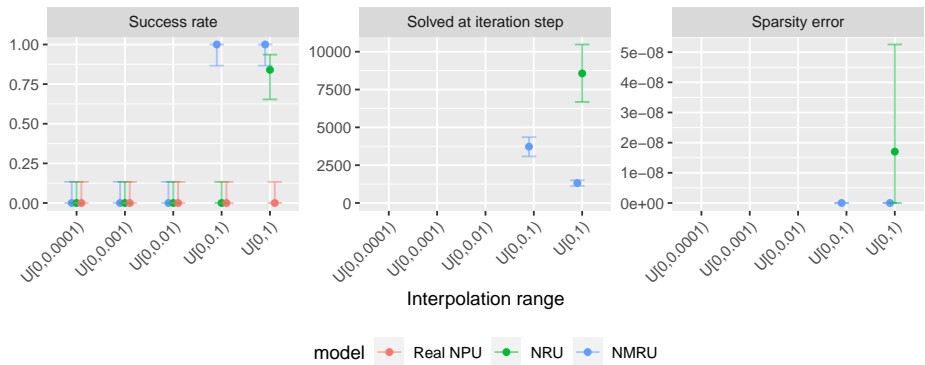

Figure 4: Input: [a,b], output $\frac{a}{b}$. Learns division of the first and second value when there is no redundancy.

## G   Real NPU; Single Module Task (with Redundancy): Additional Experiments

Figure 5 shows results of using the NPU for the task with redundancy.

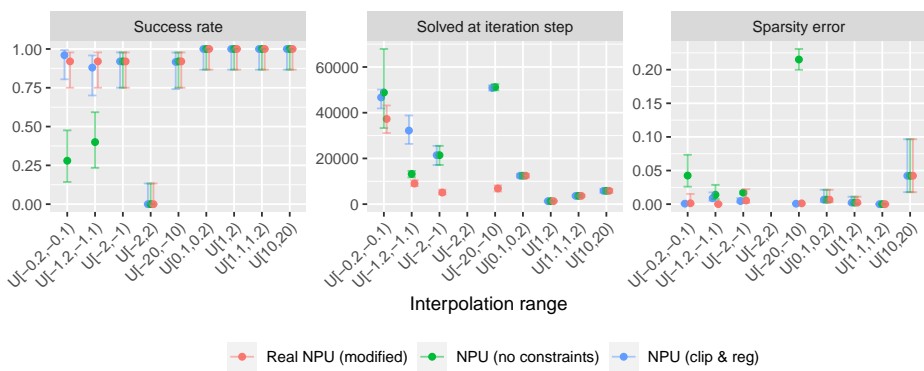

Figure 5: Adapting the Real NPU to use complex weights (NPU) on the Single Module Task with redundancy. Compares the NPU architecture with the Real NPU modifications (i.e. NPU (no constraints)) and the same model but with the imaginary weights clipped to [-1,1] and L1 sparsity regularisation on the complex weights (i.e. NPU (clip & reg)).

Figure 6 shows how modifying the weight discretisation to not penalise weights at 0 does not effect success.

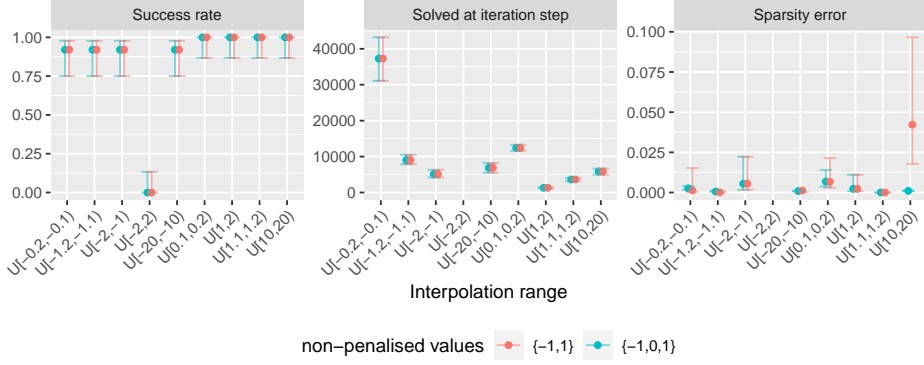

Figure 6: Comparing weight discretisation on the NPU weights which penalises not having weight of $\{-1, 1\}$ vs $\{-1, 0, 1\}$.

## H NMRU; Single Module Task with Redundancy (Additional Experiments)

This section further explores the NMRU architecture.

Figure 7 shows an ablation study on different components of the NMRU architecture. Removing both the sign retrieval and grad norm clipping performs poorly over a majority of ranges (including positive ranges). Gradient norm clipping alone is unable to solve the issue in learning negative ranges, however

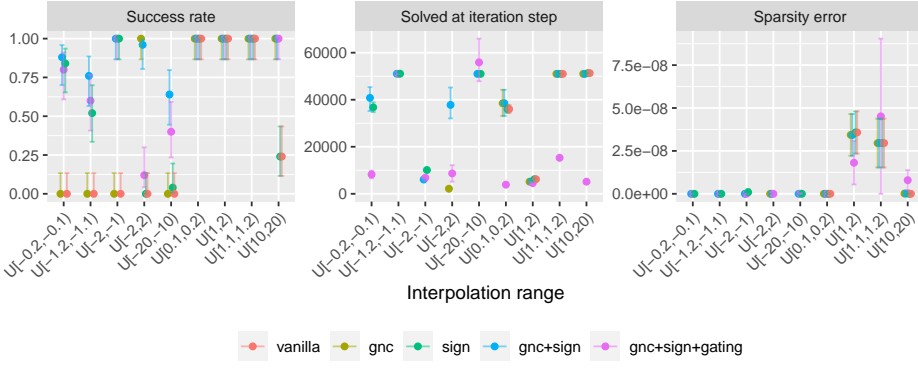

Figure 7: Ablation study for the NMRU.

fully succeeds on the $\mathcal{U}$[-2,2) range. Using the sign retrieval without the gradient clipping gains successes for the negative ranges, though performance on $\mathcal{U}$[2,-2) is effected. However, including both gradient clipping and sign retrieval results in separating the calculation of the magnitude of the output and its sign while having reasonable gradients, gaining the most improvement over the vanilla NMRU. Further including a learnable gate vector (like the Real NPU), which is applied to the input vector, hinders performance. The largest solved at iteration step seems to be bounded at approximately 50,000 iterations which correlates to the point at which the sparsity regularisation begins, which highlights the importance of discritisation. Even with the different ablations, the sparsity errors of the successful seeds remain extremely low (which is not always the case for the Real NPU (see Figure 6)).

Figure 8 shows the effect of using different learning rates on the NMRU (with grad norm clipping and sign retrieval) using an Adam optimiser. Too low a learning rate struggles on the mixed-sign range $\mathcal{U}$[-2,2). Too high a learning leads to no success on multiple ranges.

Figure 9 compares training the NMRU with either an Adam and SGD optimiser. As expected, Adam outperforms SGD in all ranges (except two, where both perform equally). This difference in performance can be accounted for by Adam's ability to scale the step size of each weight, which can compliment the clipped gradient norm of the NMRU, in contrast to the SGD's global step size.

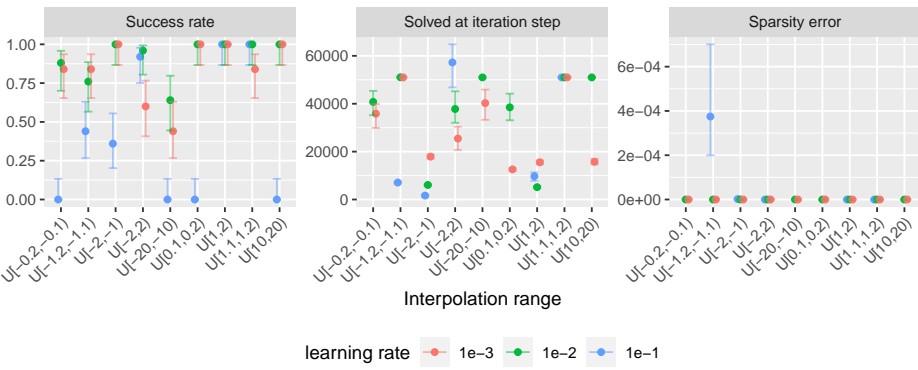

Figure 8: Effect of different learning rates on the NMRU

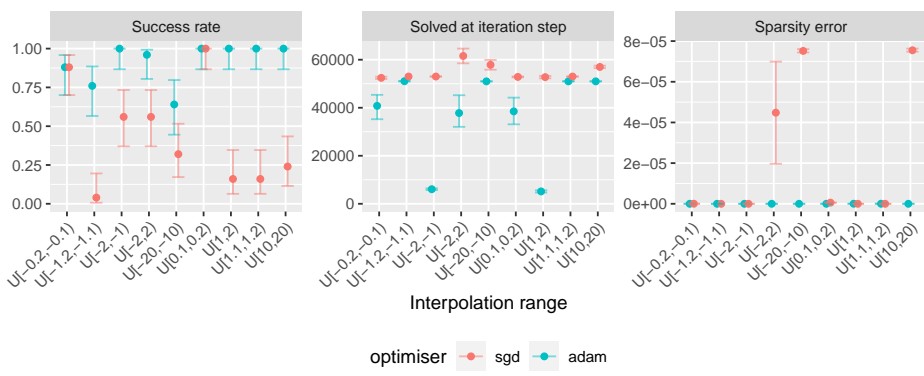

Figure 9: Effect of optimiser on the NMRU. SGD = Stochastic Gradient Descent.

## I   NRU; Single Module Task (with Redundancy): Calculating the Sign Separately

The 'separate NRU' module calculates the magnitude and sign separately and then combines them using multiplication together once all input elements are accounted for. The following definition is used to calculate a NRU with separate magnitude and sign calculation,

$$z_o = \prod_{i=1}^{I} \left( |x_i|^{W_{i,o}} \cdot |W_{i,o}| + 1 - |W_{i,o}| \right) \cdot \prod_{i=1}^{I} \text{sign}(x_i)^{\text{round}(W_{i,o})} . \tag{5}$$

Figure 10 shows results, where the separate sign method shows no difference in success to the original NRU architecture.

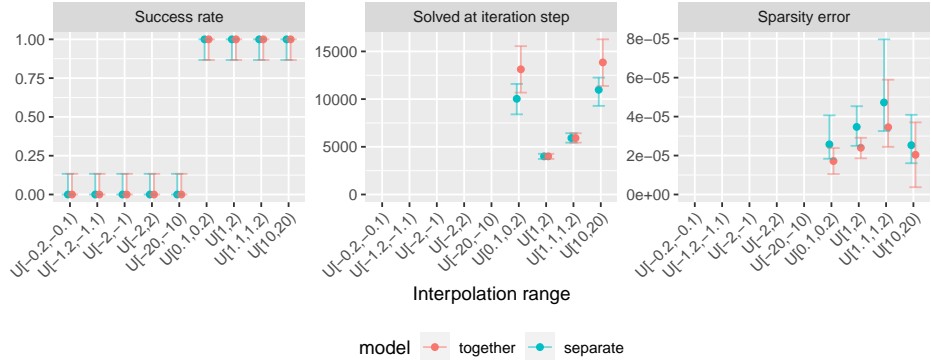

Figure 10: NRU on the redundancy experiment comparing a module which calculates the magnitude and sign together vs calculating the magnitude and sign separately and then combining them.

## J  Effect of Different Losses on the Single Module Task (with Redundancy)

Table 5: The properties of different loss functions.

|  | MSE | PCC | MAPE |
|---|:---:|:---:|:---:|
| Batch mean | ✓ | ✓ | ✓ |
| Standardisation |  | ✓ | ✓ |
| Difference of prediction from target | ✓ |  | ✓ |
| Projection |  | ✓ |  |
| Mean centering |  | ✓ |  |

Different losses induce different loss landscapes impacting the areas of success for a module. We explore the effects of three different losses including the MSE, Pearson's Correlation Coefficient (Equation 7), and the Mean Absolute Precision Error (Equation 8). We use the division task with 10 inputs. The properties of each loss is summarised in Table 5. All experiment parameters match the original MSE runs in the main experiments. The only difference is the loss used.

$$v_{x,i} = (\hat{y}_i - \bar{\hat{y}}), \quad s_x = \sqrt{\text{clamp}(\frac{1}{N} \sum_i^N v_{x,i}^2, \epsilon)}$$

$$v_{y,i} = (y_i - \bar{y}), \quad s_y = \sqrt{\text{clamp}(\frac{1}{N} \sum_i^N v_{y,i}^2, \epsilon)} \tag{6}$$

$$r = \frac{1}{N} \sum_i^N (\frac{v_{x,i}}{s_x + \epsilon} \cdot \frac{v_{y,i}}{s_y + \epsilon})$$

$$\text{pcc loss} := 1 - r \tag{7}$$

where N is the batch size, and the means ($\bar{\hat{y}}$ and $\bar{y}$) are taken over the batch. $\epsilon$ is used to provide better numerical stability.

$$\text{mape loss} := \frac{1}{N} \sum_i^N (\frac{|y_i - \hat{y}_i|}{y_i}) \tag{8}$$

**Real NPU (Figure 11)**  Both the Real NPU and MAPE are able to get success on the $\mathcal{U}$[-2,2] range, which the MSE completely fails on, implying that having a loss with standardisation is useful. However, in order to gain successes in the mixed-sign range, the other negative ranges have reduced

in success for both PCC and MAPE. Both speed and sparsity retain similar performance to MSE in a majority of cases, with PCC solving especially fast for all tested ranges.

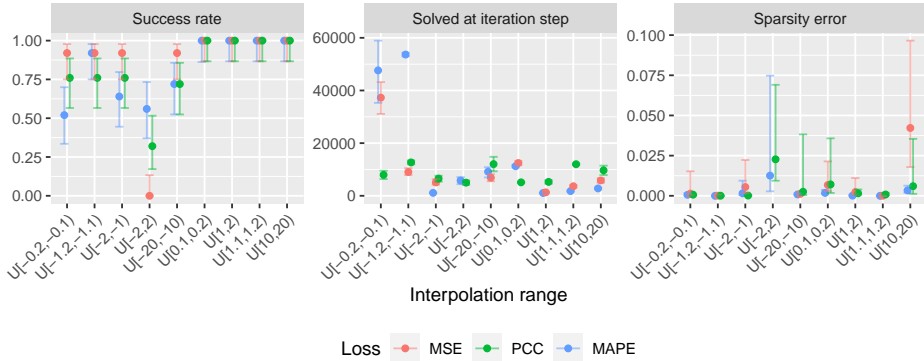

Figure 11: Single Module Task with redundancy on the Real NPU, comparing different loss functions.

154

**NRU (Figure 12)**   Different losses have little effect on the NRU. All three losses perform well on the positive ranges. Compared to the Real NPU, the PCC loss on the NRU takes longer to converge to a success for negative ranges.

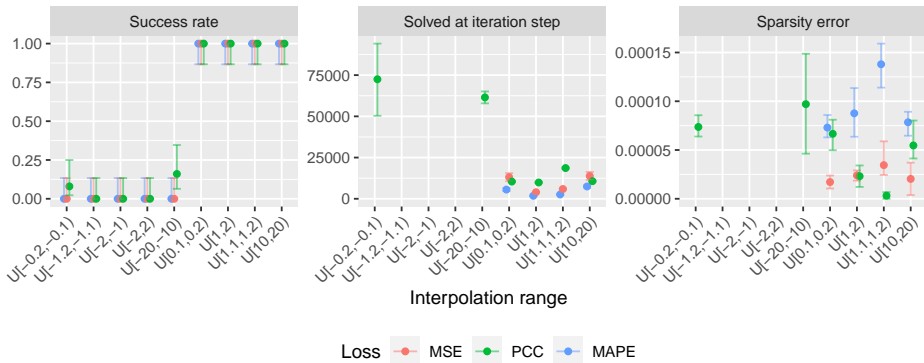

Figure 12: Single Module Task with redundancy on the NRU, comparing different loss functions.

157

**NMRU (Figure 13)**   All three loses perform reasonably well, with the PCC struggling the most. Unlike the other units, $\mathcal{U}$[-20,-10) causes the most trouble, whereas $\mathcal{U}$[-2,2) gains near to full success on two of the three losses.

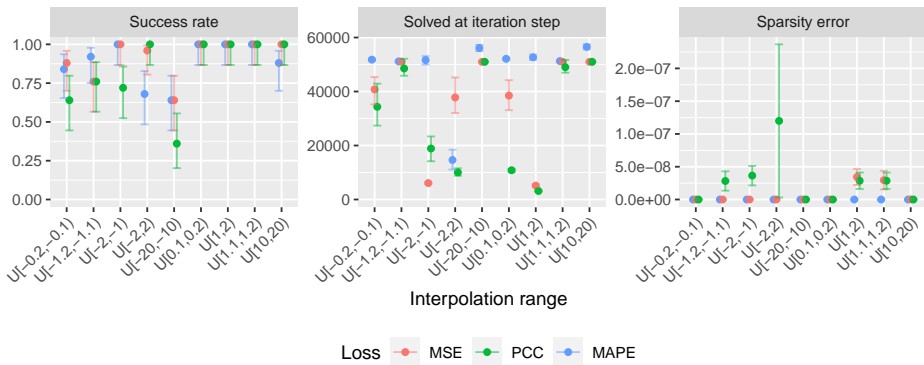

Figure 13: Single Module Task with redundancy on the NMRU, comparing different loss functions.

## K   RMSE Loss Landscapes

For clarity, we show bigger versions of each subplot from Figure 7.

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

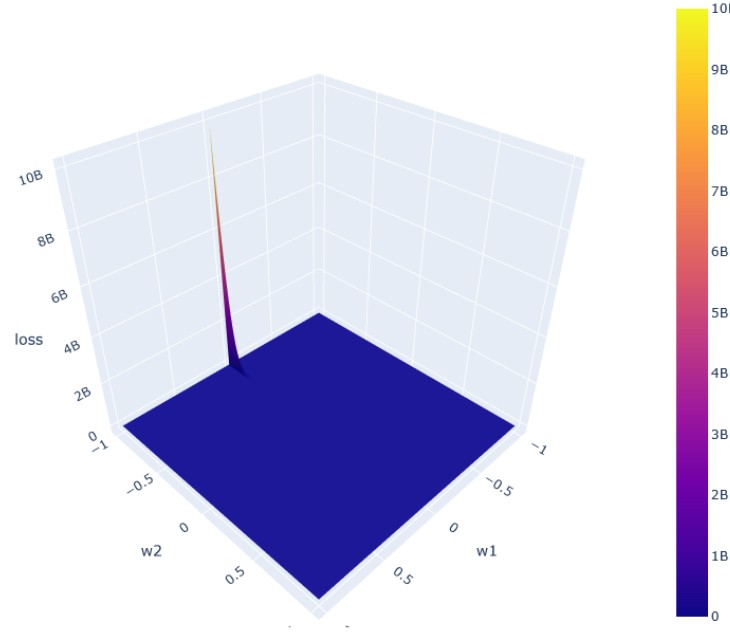

(a) NAU-Real NPU (where $\epsilon = 1e-5$)

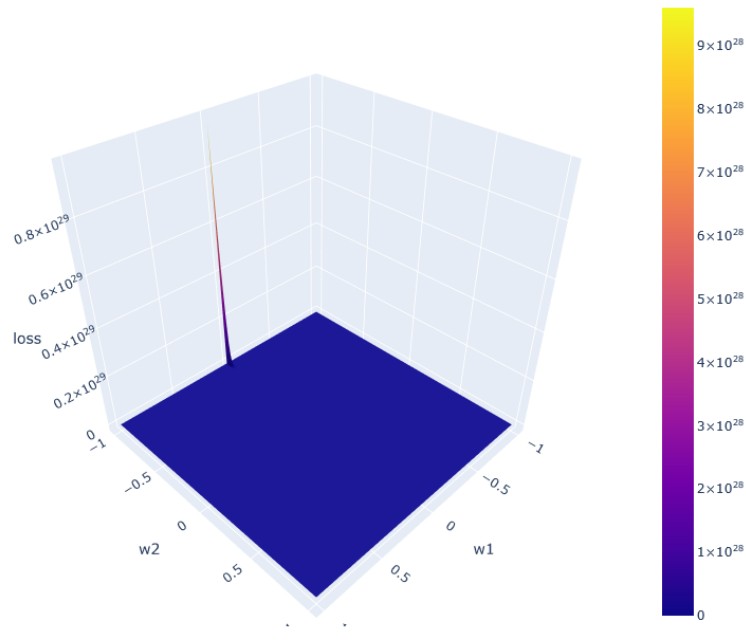

(b) NAU-NRU

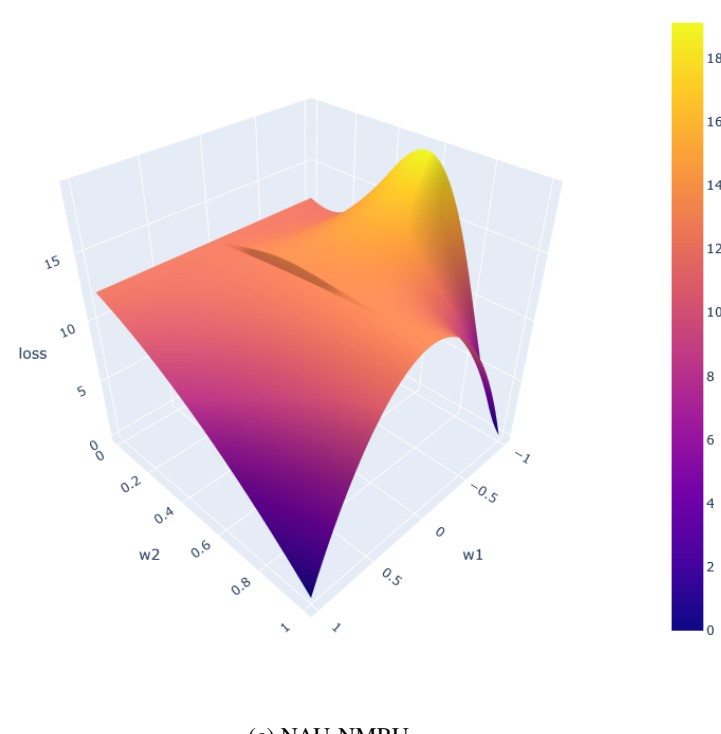

(c) NAU-NMRU

Figure 14: Enlarged loss landscapes of different stacked summative-multiplicative units.