# OpenReview forum: "Learning Division with Neural Arithmetic Logic Modules"
_NeurIPS.cc/2021/Conference — NeurIPS 2021 Submitted_

### Official Review · Reviewer_ZiPP · 2021-06-29

**Rating:** 6
**Confidence:** 1

**Summary:**

The paper is an addition to the line of study concerning neural arithmetic. Several new architectures are proposed to tackle the problem of learning the division operation. Out of the four arithmetic operations division is arguably the more challenging one and previous works have struggled with it.

Through several experimental results, the new architectures are shown the outperform previous designs, both in training accuracy and in other desirable properties

**Limitations And Societal Impact:**

 The authors adequately addressed the limitations and potential negative societal impact of their work.

**Main Review:**

Firstly, I should mention that I am a non-expert in this area. After reviewing the relevant literature I find the topic of the paper interesting and relevant. Overall, the paper is well written and presents the motivations in a compelling way.

However, there are two issues which I would like to raise:

1. The ideas underlying the new architectures seem to be a bit incremental with respect to previous works. While it is definitely a serious paper that merits publication,  I am just not 100% convinced that NeurIPS is the correct venue.

2. While the authors support their choice of architecture with a plethora of experiments, I feel that the paper lacks some theoretical foundations, which do appear to some extent in previous works. This makes it hard to interpret some of the experimental results. For example (but definitely not the only one), in Figure 3 why should I expect to have different behaviors between different interpolation ranges. How can one explain the fact that the real NPU is competitive with the new architectures on some ranges, and why those specific ranges?
Moreover, as the authors note, the most challenging ranges are those that include 0. However, only one such range is tested ([-2,2]). I would expect to have more such ranges, both in different scales and with different levels of symmetry/asymmetry.

Minor comments:
- Display (1): Perhaps it is better to mention that the multiplication is the Hadamard (or Schur) product.

**Time Spent Reviewing:**

6

---

> ### Author Response · Authors · 2021-08-09
> **Response to reviewer ZiPP - #1**
>
> Thank you for your review.
>
> The ideas underlying the new architectures seem to be a bit incremental with respect to previous works.
> - The aim of this work is not to produce a novel state-of-the-art architecture. Rather, our aim is to discover what does/does not work regarding NALMs for division, analyse the causes of failures and improve upon previous limitations. We believe that this broader analytical approach, in contrast to developing a single module approach, will benefit the field with respect to understanding NALMs.
>
> I am just not 100% convinced that NeurIPS is the correct venue.
> - As explained to reviewer mB6g, previous works in this field, have been published in conferences including NeurIPS (i.e., the NPU [5] and the NALU [6]) and ICLR (the NAU [7]) which is an equally reputable conference. We believe that publishing to NeurIPS is therefore relevant and can aid in providing further exposure for this area and to the researching community.
>
> I feel that the paper lacks some theoretical foundations, which do appear to some extent in previous works.
> - Can you elaborate on examples of what you mean by ‘theoretical foundations’? The new modules that we formulate to address limitations found from this empirical study are theoretically grounded in the sense that they provably have weight configurations that allow them to solve the task at hand. Proving learnability of those weights is however very difficult, so instead, we show this empirically (and highlight where weights are still hard to learn).
>
> paper lacks some theoretical foundations … makes it hard to interpret some of the experimental results. For example (but definitely not the only one), in Figure 3 why should I expect to have different behaviours between different interpolation ranges. How can one explain the fact that the real NPU is competitive with the new architectures on some ranges, and why those specific ranges?
> - The different behaviours of different ranges can be largely explained by two factors. Firstly, the modules are trained on the unnormalised inputs meaning the input values have a significant effect on the learning dynamics e.g., gradients or loss. Secondly, the architectures of the modules themselves vary which affect the way in which a solution is learnt. For example, the modifications made on the RealNPU show how utilising a mechanism like clipping can aid in performance, or the supplementary material Appendix H shows an ablation study on the NMRU explaining the effect of each component of the unit.
>
> Moreover, as the authors note, the most challenging ranges are those that include 0. However, only one such range is tested ([-2,2]). I would expect to have more such ranges, both in different scales and with different levels of symmetry/asymmetry.
> - We agree that the use of alternate distributions such as those with large scales (e.g., exponential/logarithmic) would cause modules to show failure cases. As per reviewer mhZR’s suggestion, we can include experiments on Benford’s distribution. Furthermore, we can also add results which other distributions which incorporate 0. However, we like to highlight that the goal of this paper is to better understand the properties of division modules that cause success/failure.
>
> Display (1): Perhaps it is better to mention that the multiplication is the Hadamard (or Schur) product.
> - We will define this operation in the paper revision.

---

> > ### Comment · Reviewer_ZiPP · 2021-08-23
> > **Response to Rebuttal**
> >
> > Thank you very much for the clarifications.
> >
> > After reviewing everything carefully I have decided to keep my score.
> >
> > - As explained to reviewer mB6g, previous works in this field, have been published in conferences including NeurIPS (i.e., the NPU [5] and the NALU [6]) and ICLR (the NAU [7]) which is an equally reputable conference. We believe that publishing to NeurIPS is therefore relevant and can aid in providing further exposure for this area and to the researching community.
> >
> > I definitely agree that, as evidenced by the previous papers, the topic is relevant for NeurIPS. My point was that the current work seems incremental with respect to those papers. I am aware that this point is very subjective, but as reviewers, we need to make such decisions.

---

> > > ### Author Response · Authors · 2021-08-26
> > > **Response to reviewer ZiPP - #2**
> > >
> > > Thank you for the clarification of your concern. We would like to clarify the motivations for the approach to our work. Our additional empirical analysis on existing modules allows us to identify areas of failure and determine ways in which we can alleviate these failures and validate them empirically.  We believe our contributions should not be viewed as incremental changes to existing modules but a result of a much more rigorous analysis, provided by our range of experimental results and theoretical findings. For example:
> > > - We are the first to provide insight for the Real NPU’s lack of robustness against a range of different training distributions (Figures 3 and 6), finding issues in particular relating to having mixed-signed inputs (Figure 4) and a stress test of its sensitivity when learning with data towards the singularity point of 0 (Figure 5, Figure 7 and Appendix F).
> > > - Through rigorous empirical studies, we show how intuitive architectural additions on the Real NPU can lead to a significant improvement to its robustness (Section 5, Figure 3 and Figure 6).
> > > - We discover the existence of modules learning to exploit multiplicative rules leading to biases in the type of solutions discovered (section 6.3.2) and further show for the Real NPU how this can be alleviated (Appendix G).
> > > - Using what we learn about from the Real NPU, we create the two novel modules the NRU and NMRU and provide substantial empirical results to conclude the trade-offs for using the different modules in different situations.
> > > - Similar to the iterative approach used on the Real NPU in section 5, we provide an ablation study in Appendix H on the NMRU providing evidence to justify its design decisions.
> > > - To further explore the sensitivity of modules, we are the first to provide results on the effect of two additional loss functions (PCC and MAPE) in Appendix J in which we discover additional trade-offs.
> > > - Using the accumulated knowledge from the work in this paper, we provide a desiderata for division modules, detailing the theoretic properties which a module should adhere to in Appendix A.
> > >
> > > Hence, our contributions should not be seen as incremental but a necessary step in understanding the learning dynamics of these modules such that researchers can better utilize them in future work.

---

> > > > ### Comment · Reviewer_ZiPP · 2021-08-31
> > > > **Response**
> > > >
> > > > Again, thank you.
> > > >
> > > > I sincerely appreciate the explanations and the added experiments.
> > > >
> > > > - I am a bit worried that I cannot see the results in an organized fashion. However, this is the (unfortunate) standard put forth by the conference and I do not see a reason why you should be penalized for it.
> > > >
> > > > - As I indicated in my initial review, this is not my area of expertise. So, paraphrasing the wording of the conference, my initial "assessment is an educated guess". While I am still not 100% convinced that my initial assessment is off, your well-constructed response has left me willing to reconsider my position.
> > > >
> > > > In light of the above, I have decided to raise my score to 6.
> > > >
> > > > Good luck.

---

### Official Review · Reviewer_mB6g · 2021-07-12

**Rating:** 5
**Confidence:** 3

**Summary:**

In this paper, the authors explore neural arithmetic architectures that could learn to perform division tasks, both in the case of input redundancy as well as with only 2 operators. Besides improving an existing architecture (NPU), other two related architectures are introduced (NRU and NMRU), both based on the idea that division can be framed as a multiplication of reciprocals. Extrapolation capabilities are probed by considering testing intervals that do not overlap with training intervals. The newly introduced architectures achieve better performance, though in the case of input redundancy all models are still error prone.

**Ethical Concerns:**

I do not see ethical concerns.

**Limitations And Societal Impact:**

The authors properly discuss the limitations of their research approach. The potential negative societal impact discussed by the authors seems quite speculative.

**Main Review:**

Originality: This research work appears mostly incremental in nature. It extends and improves previous model architectures, which are empirically tested over previously established tasks.

Quality: The research topic is very actual, given the recent interest in testing mathematical learning skills of deep nets, though the scope of the work appears somewhat limited given the specificity of the research approach. The applicability of these models to broader contexts / domains is not particularly evident. The paper builds on the methodological setup established in previous work, which appears sound and well-designed. As a possible improvement, given that the proposed NMRU achieves reasonable accuracy over all tested ranges it would be interesting to further test its extrapolation capabilities by including even more challenging numerical ranges.
Overall, my feeling is that this is a well-conducted research work, but probably NeurIPS might not be the most appropriate venue to present these findings.

Clarity: The paper is well written and properly organized. I found the opening sentence a bit convoluted: I think there could be more effective ways to introduce to the reader the challenging aspects of the division problem. Moreover, the authors do not  clearly explain why we should initially focus on the problem of learning division when there is input redundancy.

Significance: The issue of learning mathematical knowledge with deep nets is timely and challenging, and the deep learning community is spending increasing efforts in this direction. However, the significance of this paper for the broader NeurIPS community might be limited, because the current setup is very specific. The “broader impact” proposed by the authors (e.g., ability to produce transparent generalizable solutions) appears quite speculative and should be more clearly demonstrated.


**Time Spent Reviewing:**

5

---

> ### Author Response · Authors · 2021-08-09
> **Response to reviewer mB6g - #1**
>
> Thank you for your review.
>
> The applicability of these models to broader contexts / domains is not particularly evident.
> - Even though the field of NALMs is relatively new, there are applications that use NALMs in larger end-to-end networks. For example: using a gated summative and multiplicative NALM for trajectory prediction on road sections [1], using a summative NALM to extract spatial/temporal features for fast badminton stroke classification [2], or using NALMs to help schedule views on content-delivery-networks for crowdsourced-live-streaming [3]. Furthermore, the architectures of NALMs have been applied to other domains such as the Neural Multiplication Unit, inspiring a formulation for expressing the conjunction of binary detectors which express interpretable rules in the dynamic microbiome domain [4].
>
> As a possible improvement, given that the proposed NMRU achieves reasonable accuracy over all tested ranges it would be interesting to further test its extrapolation capabilities by including even more challenging numerical ranges.
> - A similar point is also made by reviewer mhZR. We decided to focus on the uniform distribution in the paper because a) a comparison of different ranges for the RealNPU has not been done before and b) the properties of the Uniform distribution allows for us to easily understand the properties of the underlying data (such as ranges, magnitude and variance). That being said, we can add in the revision results with a larger magnitude from a more challenging distribution e.g., exponential/logarithmic.  We believe that using such distributions will display further traits of these modules, even showing new failure cases.
>
> NeurIPS might not be the most appropriate venue to present these findings.
> - Previous works in this field, have been published in conferences including NeurIPS (i.e. the NPU [5] and the NALU [6]) and ICLR (the NAU [7]) which is an equally reputable conference. We believe that publishing to NeurIPS is therefore relevant and can aid in providing further exposure for this area and to the researching community.
>
> I found the opening sentence a bit convoluted: I think there could be more effective ways to introduce to the reader the challenging aspects of the division problem.
> - Thank you for raising the concern. We understand that the start of the introduction can feel a bit sudden. We believe reorganising the introduction paragraph, such that lines 14-17 are moved to come after the ending of the sentence in line 23 (with a slight rewording) would improve introducing the problem.
>
> Moreover, the authors do not clearly explain why we should initially focus on the problem of learning division when there is input redundancy.
> - That’s a good point. As mentioned in [7]: ‘Redundant units are very common in neural networks, which are often overparameterized’. As an application of NALMs is the ability to be integrated into larger overparametrized neural networks as an intermediate module [8], having the ability to success select the relevant inputs is important, otherwise the ability to act as interpretable modules is not met.
>
> The significance of this paper for the broader NeurIPS community might be limited, because the current setup is very specific.
> - The scope of this paper is to provide insights for the reader regarding the learning of different interpretable modules for division. By showing that difficulties exist even on a simple task with synthetic data, and beginning to highlight ranges of data and architecture choices (see supplementary material Appendix A) which influence performance, we raise awareness for researchers to consider in their work.  Ideally, this paper will act as a stepping stone for researchers who may develop modules which require having similar properties such as: discretised weights, input redundancy, or robustness to unnormalized inputs. Furthermore, this line of research falls under the broader category of interpretability, which is becoming increasingly important when considering the use of neural networks for real world applications.
>
> The “broader impact” proposed by the authors (e.g., ability to produce transparent generalizable solutions) appears quite speculative and should be more clearly demonstrated.
> - To clarify the broader impact: NALMs are designed such that by observing its weights we can understand that underlying arithmetic relation between the input and output it is trained on. Alone, these modules have limited expressability. However, as the learning is differentiable and only requires specifying an input and output size, we are able to use NALMs as a sub-module in a large end-to-end network, which may contain complex networks such as LSTMs, CNNs, MLPs, Transformers etc.  For further information regarding such applicability of the modules, see section 7 ‘Applications in NALU’ from [8].
>
> The potential negative societal impact discussed by the authors seems quite speculative.
> - This paper focuses on the lower levels of the impact stack (see https://medium.com/@GovAI/a-guide-to-writing-the-neurips-impact-statement-4293b723f832). Therefore, in its current stage of research we can only speculate the negative impacts which may come from this work. If the reviewer has concrete suggestions, we would happily take them on-board however.
>
> **References:**
>
> [1] Xiao, Z., Li, F., Wu, R., Jiang, H., Hu, Y., Ren, J., Cai, C. and Iyengar, A., 2020. Trajdata: On vehicle trajectory collection with commodity plug-and-play obu devices. IEEE Internet of Things Journal, 7(9), pp.9066-9079.
>
> [2] Raj, A., Consul, P. and Pal, S.K., 2020, September. Fast Neural Accumulator (NAC) Based Badminton Video Action Classification. In Proceedings of SAI Intelligent Systems Conference (pp. 452-467). Springer, Cham.
>
> [3] Zhang, R.X., Ma, M., Huang, T., Pang, H., Yao, X., Wu, C., Liu, J. and Sun, L., 2019, October. Livesmart: A qos-guaranteed cost-minimum framework of viewer scheduling for crowdsourced live streaming. In Proceedings of the 27th ACM International Conference on Multimedia (pp. 420-428).
>
> [4] Maringanti, V.S., Bucci, V. and Gerber, G.K., 2020. Scalable learning of interpretable rules for the dynamic microbiome domain. bioRxiv.
>
> [5] Heim, N., Pevny, T., and Smidl, V. 2020. Neural Power Units. In Advances in Neural Information Processing Systems (pp. 6573–6583). Curran Associates, Inc..
>
> [6] Trask, A., Hill, F., Reed, S., Rae, J., Dyer, C., and Blunsom, P. 2018. Neural Arithmetic Logic Units. In Advances in Neural Information Processing Systems. Curran Associates, Inc..
>
> [7] Andreas Madsen, and Alexander Rosenberg Johansen 2020. Neural Arithmetic Units. In International Conference on Learning Representations.
>
> [8] Mistry, B., Farrahi, K. and Hare, J., 2021. A Primer for Neural Arithmetic Logic Modules. arXiv preprint arXiv:2101.09530.

---

### Official Review · Reviewer_H6xb · 2021-07-16

**Rating:** 3
**Confidence:** 5

**Summary:**

This manuscript proposed new types of neural arithmetic modules called Neural Reciprocal Units (NRU) and Neural Multiplicative Reciprocal Units (NMRU). NRU and NMRU extend Neural Multiplication Units by applying power terms and reciprocal inputs, respectively, and thus they enable divide operation. They also proposed improvement method of Neural Power Units (NPU).

**Main Review:**

This manuscript introduces novel arithmetic layers, which enables divide operation. Proposed NRU and NMRU achieved higher success rates and fast learning ability compared to Real NPU in various experiments.
The manuscript seems to cover too broad subject. Proposing an improvement method of NPU does not seemed to be highly related to NRU and NMRU. It would be better to add comments explain relationship among them, or focusing more on NRU and NMRU. Additionally, there is a concern about novelty on the proposed improvement method of NPU, since they are briefly commented on ‘Neural Power Units’(NIPS 2020).
In Equation (5), they used summation of cosine function to calculate sign value of output, but I think it should be product of cosine like Equation (5) in supplementary material.

**Time Spent Reviewing:**

6 hours

---

> ### Author Response · Authors · 2021-08-09
> **Response to reviewer H6xb - #1**
>
> Thank you for your review.
>
> The manuscript seems to cover too broad subject.
> - We believe the contrary to be true. Our focus in the paper is only for the division operation (which is considered the most difficult of the 4 main arithmetic operations to learn). Furthermore, we specifically focus on NALMs which are an upcoming field of research. Our experiments are designed to be specific to division of numbers in the simplest sense (i.e. dividing two numbers or selecting two numbers then dividing), meaning we are able to find the direct causes of failure and success for these modules.
>
> Proposing an improvement method of NPU does not seemed to be highly related to NRU and NMRU. It would be better to add comments explain relationship among them, or focusing more on NRU and NMRU.
> - We believe this is untrue. The design of the NMRU takes direct influence from the RealNPU in the way we calculate the sign.  Furthermore, both the NRU and NMRU use clipping, discretisation regularisation, and the constrained Xavier Uniform initialisation as relevant for each module. For a more detailed cross-comparison of the architectures see in the supplementary material Appendix A (Properties of a Division Module) where we compare the similarities and differences of the modules against such categories.
>
> There is a concern about novelty on the proposed improvement method of NPU, since they are briefly commented on ‘Neural Power Units’(NIPS 2020).
> - Our focus includes taking the existing work surrounding NALMs for division and provide improved understanding. We introduce two novel modules (the NRU and NMRU) and provide new understanding into types of inputs which cause robustness difficulties and properties of the module architecture’s which aid with improving robustness (e.g. a sign mechanism or discretisation regularisation).  For the NPU, we purposely show the effects of adding changes to the original RealNPU architecture rather than completely changing the original.  If this does not answer your question, can you elaborate on your ‘concern about novelty’?
>
> In Equation (5), they used summation of cosine function to calculate sign value of output, but I think it should be product of cosine like Equation (5) in supplementary material.
> - Equation 5 does require a summation when calculating the cosine portion of the equation. Intuitively, the expression: ‘sum(cos(W_i,o * k_i)’ represents a matrix multiplication but in element form. The equation has a similar functionality to the use of cos in the RealNPU (equation 2) with the difference being equation 2 is written in matrix form rather than in an element wise form.

---

> > ### Comment · Reviewer_H6xb · 2021-08-24
> > **Response to rebuttal**
> >
> > Thanks the authors for the explanations.
> > - My concerns mainly focus on chapter 5 (improvement of NPU). Authors have suggested 4 ways of improvement, regularization, clipping, enforcing discretization and initialization, but it seemed that applying regularization, clipping and initialization are trivial methods. Especially, L1-regularization is already applied to NPU (eq 18) in ‘Neural Power Units’(NIPS 2020).
> > - Equation 5 is not clear to me. If I understand correctly, when the network is trained to multiply all x_i, then W_1,o, W_2,o, … W_i,o will result in 1 and W_i+1, W_i+2, … W_2i will be 0, resulting in z_o = abs(x_0*x_1*…x_i) * (cos(k_1)+cos(k_2)+…cos(k_i)). This equation can be different from x_1*x_2*…x_i, target of network because cos(k_1)+cos(k_2)+…cos(k_i) is not restriected to 1 or -1, but can be varied from –i to i. I think cos(sum(W_i,o+k_i)) or multiply(cos(W_i,o+k_i)) can be sign of multiplication as NPU.

---

> > > ### Author Response · Authors · 2021-08-25
> > > **Response to reviewer H6xb - #2**
> > >
> > > My concerns mainly focus on chapter 5 (improvement of NPU). Authors have suggested 4 ways of improvement, regularization, clipping, enforcing discretization and initialization, but it seemed that applying regularization, clipping and initialization are trivial methods. Especially, L1-regularization is already applied to NPU (eq 18) in ‘Neural Power Units’(NIPS 2020).
> > > -	Thank you for raising your concerns. The wider contributions of the paper are to analyze the loss landscape/learning dynamics and to explore for a module what does and does not work in order to get good extrapolation. Therefore, the suggested improvements are to show the effect of how small changes can improve the robustness against the original module which we found struggles with different input ranges even when there are no redundant inputs (see Figure 3). The fact that these changes seem to work well in comparison to not using them should raise awareness that such methods can be utilised by future research/development of new modules. We are aware that the L1 regularisation already is used in the NPU paper, which is why we question if it is required, as this remains unanswered in the original NPU paper. We ask this because intuitively, using a regulariser which encourages sparsity may be counter productive as there can be cases when we want all inputs to be selected and therefore not be 0.
> > >
> > > Equation 5 is not clear to me. If I understand correctly, when the network is trained to multiply all x_i, then W_1,o, W_2,o, … W_i,o will result in 1 and W_i+1, W_i+2, … W_2i will be 0, resulting in z_o = abs(x_0x_1…x_i) * (cos(k_1)+cos(k_2)+…cos(k_i)). This equation can be different from x_1x_2…x_i, target of network because cos(k_1)+cos(k_2)+…cos(k_i) is not restriected to 1 or -1, but can be varied from –i to i. I think cos(sum(W_i,o+k_i)) or multiply(cos(W_i,o+k_i)) can be sign of multiplication as NPU.
> > > -	Apologies, we misinterpreted your previous comment about equation 5. You are correct, there is a typo. The 2nd part of equation 5 should be a sum and be inside the cos, i.e., $cos(\sum_{i=1}^{2I}(W_i,o \cdot x_i))$.  Thank you for bringing this back to our attention again, we shall amend the typo. Furthermore, as you suggested, using a product instead i.e.$ \prod_{i=1}^{2I}(cos(W_i,o \cdot x_i))$ is equally valid. We use the sum notation in equation 5 because it reflects how the module is implemented, by using efficient vectorization for batched data.
> > >  - For full clarity, allow us to work through an example using equation 5. Assume we are given 3 inputs ($x_1, x_2, x_3$) and we want to multiply all of them to get $x_1\cdot x_2 \cdot x_3$ (as suggested in your previous comment). Furthermore, let $x_2$ be negative so the output will be negative. $2I = 6$ represents the 3 inputs concatenated with their reciprocals (i.e. $x_1, x_2, x_3, x_4, x_5, x_6$ where $x_4=\frac{1}{x_1}, x_5=\frac{1}{x_2}$ and $x_6=\frac{1}{x_3}$). To get the desired output  $x_1\cdot x_2 \cdot x_3$ the weights corresponding to each input require to be 1’s for the first half of the input and 0’s for the second half, i.e., $W_1,o = W_2,o = W_3,o = 1$ and $W_4,o = W_5,o = W_6,o = 0$. Now consider the first part of equation 5. A weight of 1 will select the corresponding input value and a weight of 0 will convert the input to 1 (which results in the input having no effect on the final output). So applying the given weights will result in $|x_1| \cdot |x_2| \cdot |x_3|$. Now we have the magnitude of the output (from using part 1 of eq 5), and all that’s left is to calculate the sign (part 2 of eq 5), i.e., $cos(\sum_{i=1}^{2I}(W_i,o \cdot x_i))$. $k_i$ will be $\pi$ if the input is negative and $0$ otherwise. The resulting expression will be: $cos( (1\cdot 0) + (1 \cdot \pi) + (1 \cdot 0) + (0 \cdot 0) + (0\cdot \pi) + (0 \cdot 0) ) = cos(0+ \pi + 0 + 0 + 0 + 0) = cos(\pi) = -1$. Because of the cyclic nature of cos, whenever a negative input is being processed (and the input is relevant i.e. weight of 1) the result of $W_i,o \cdot k_i = \pi$. So each time you process a negative input you accumulate by $\pi$. Every even multiple of $\pi$ when passed through cos (e.g. $cos(2\pi), cos(4\pi))$ will evaluate to 1 and every odd multiple will evaluate to -1 allowing for the appropriate resultant sign value.

---

### Official Review · Reviewer_mhZR · 2021-07-16

**Rating:** 7
**Confidence:** 3

**Summary:**

This paper study possible adaptations of Neural Arithmetic Linear Units (Trask 2018) to better handle division. The authors first propose improvements for training Real Neural Power Unit (Heim 2020), a neural arithmetic unit which performs division as substraction in log-space, using cosines to handle signs, and a gating mechanism to select operands, while constraining all weights to remain close to 0, 1 or -1. They show that introducing L1 regularisation, gradient clipping, and tuning weight constraint and initialization greatly improve its accuracy when dividing numbers.

They introduce two new designs, the Neural Reciprocal Unit, which functions as an addition in log-space with weights constrained to -1, 0 and 1, and can therefore perform multiplication or division of numbers (and "unselect" them when weights are zero, which dispenses of the gating mechanism), and the Neural Reciprocal Modified Unit, where the reciprocals are provided as additional inputs, constraining weights to 0 and 1 (as a selection mechanism), and adds a cosine function to handle signs.

Testing over a variety of generated samples, they show that whereas the new designs improve on the original Real NPU, most of the still struggle on input of mixed signs, and have difficulty working with numbers close to zero, and learning both selection (choosing the operands) and division.

**Limitations And Societal Impact:**

They have

**Main Review:**

**Originality**
The paper builds on existing designs and experiments. The techniques introduced in section 5 to improve performance of Real NPU are standard. The new designs proposed are incremental or crossovers between existing NALU modules.

**Clarity**
The paper is very technical, but well written, and provides a lot of details about proposed architectures and experiments.

In section 3.1, it would be useful to define  the symbol used for element-wise multiplication. Also, the notation W^r W^i for the real and imaginary weight matrice conflict with the r symbol used for the gate. The ordering of the interpolation datasets in figure 3 and 6 is counter-intuitive: decreasing order of negative numbers, then increasing of positive)

**Quality**
Very detailed and interesting experiments. The results are extremely convincing, and the experiments are the strong point of this paper.

A few questions and comments :

1- In section 3.2: is the use of tanh to deal with the local discontinuity of the derivative of absolute value really necessary? RELU and absolute value activation functions are routinely used in feedforward networks, despite having the same gradient discontinuity.

2- All the data in the training and test set are uniformly sampled from relatively small intervals. This has an impact on the distribution of results. In the case of multiplication and division, it might be interesting to test over a very large interval, with a logarithmic (Benford) distribution.

3- In section 5, experiments with L1 regularization suggest that it is needed but in very small amounts. Could L2 regularization be a better choice? (as it would have the same stabilizing effect, but less enforce sparseness).

4- Being able to extrapolate out of the training distribution support is one of the main justification of NALU. In the paper, most experimental results are calculated from interpolation test set. Adding more extrapolation results would reinforce the paper.

**Significance**
The main significant result is the improvement of the Real Neural Power Unit training. NRU fail to convince because of their problem on mixed sign operands. The experiments do a good job at documenting the difficulty of training NALU on division.


**Time Spent Reviewing:**

4

---

> ### Author Response · Authors · 2021-08-09
> **Response to reviewer mhZR - #1**
>
> Thank you for your review.
>
> In section 3.1, it would be useful to define the symbol used for element-wise multiplication
> - We will update the paper to define the Hadamard product.
>
> The notation W^r W^i for the real and imaginary weight matrice conflict with the r symbol used for the gate.
> - We will update the paper, modifying the notation for W^r to W^re and W_i to W_im to avoid any confusion.
>
> The ordering of the interpolation datasets in figure 3 and 6 is counter-intuitive: decreasing order of negative numbers, then increasing of positive)
> - We will update the paper so figure 3 and 6 have an increasing order in the x-axis i.e. [-20,-10), [-2,-1), [-1.2,-1.1), [-0.2,-0.1), [-2,2), [0.1,0.2), [1.1,1.2), [1,2), [10,20).
>
> 1 - In section 3.2: is the use of tanh to deal with the local discontinuity of the derivative of absolute value really necessary? RELU and absolute value activation functions are routinely used in feedforward networks, despite having the same gradient discontinuity.
> - We believe the use of tanh is necessary. Though it is true that the use of ReLU and absolute are prevalent in Deep Neural Networks (DNNs), we would like to emphasize that such networks are highly over-parameterised meaning that they have the capacity to compensate for the lack of differentiability of such activation functions. In contrast, our focus lies on NALMs which are tightly-parameterised networks and therefore do not have the capacity to work such functions in the way DNNs can. Hence, we rely on using a tanh which can provide a gradient for the entire domain.
>
> 2 - All the data in the training and test set are uniformly sampled from relatively small intervals. This has an impact on the distribution of results. In the case of multiplication and division, it might be interesting to test over a very large interval, with a logarithmic (Benford) distribution.
> - As you’ve pointed out, using an alternative distribution along with an ‘extreme’ range will display a different story to using a uniform distribution on small intervals.  In the paper revision, we can add the results of modules trained on a Benford distribution.  To note, in additional experiments (not included in the current submission) we have found that training the modules on the redundancy experiment setting using an exponential distribution (beta value 500) finds modules to struggle with convergence more so than when using a uniform distribution on a smaller interval. In particular, the NMRU produces NANs when training. However, we do have a solution which can be applied to the NMRU to avoid this issue, which we can further discuss if desired.
>
> 3 - In section 5, experiments with L1 regularization suggest that it is needed but in very small amounts. Could L2 regularization be a better choice? (as it would have the same stabilizing effect, but less enforce sparseness).
> - Interesting suggestion. From trying this, we found that L2 performs worse than L1 (especially on the positive ranges). Most likely, the bias of L2 to force small weights can conflict with weights which are required to have values such as -1 and 1. Furthermore, just to clarify, the L1 was initially chosen (rather than L2) because it was used in the original NPU experiments [1].
>
> 4 - Being able to extrapolate out of the training distribution support is one of the main justifications of NALU. In the paper, most experimental results are calculated from interpolation test set. Adding more extrapolation results would reinforce the paper.
> - To clarify, all results are displaying the performance with respect to the extrapolation range. The reasons why the x-axis shows the interpolation range is to a) be consistent with the presentation style used in previous works [2] and b) to make it easier for the reader to easily identify the training distributions which cause difficulties in learning e.g. [-2,2]. If you still believe further extrapolation results are required, could you elaborate on what you would like to see. (Note that we are planning to update the paper with results from the Benford’s distribution as per your question 3.)
>
> **References:**
>
> [1] Heim, N., Pevny, T., and Smidl, V. 2020. Neural Power Units. In Advances in Neural Information Processing Systems (pp. 6573–6583). Curran Associates, Inc..
>
> [2] Andreas Madsen, and Alexander Rosenberg Johansen 2020. Neural Arithmetic Units. In International Conference on Learning Representations.

---

### Author Response · Authors · 2021-08-25
**Summary of findings from additional experiments - #1**

The following summaries the results from running additional experiments as suggested by reviewers.

***
**Experiment: Benford (logarithmic) distribution - testing large numbers**

Suggested by reviewer(s)/ relevant for reviewer(s):  **mhZR, ZiPP, mB6g, ZiPP**

**Summary:** Overall the Benford distribution revealed an area of challenge for the NMRU.

We train modules to divide numbers generated from the Benford distribution with training range [10,100) and test range [100,1000). Modules tested include: RealNPU (modified), NRU and NMRU. For input size 2, all modules show full success with the NMRU solving the problem fastest and the RealNPU (modified) being the slowest. For input size 10, the RealNPU (modified) and NRU have full success, while the NMRU’s success rate is 0.16.
***
**Experiment: Larger ranges (/ranges around 0)**

Suggested by reviewer(s)/ relevant for reviewer(s):  **mB6g, ZiPP**

**Summary:** Overall, all modules find the larger ranges to be challenging when redundant inputs exist.

On the 2 and 10 input setups we train modules (RealNPU (modified), NRU and NMRU) on uniform ranges: U[-100,100) and U[-50,50) with test ranges U[[-200, -100) $\cup$ [100, 200)] and U[[-100, -50) $\cup$ [50, 100)] respectively. On the 2-input setup, both NRU and SignNMRU have full success and the RealNPU (modified) has failure cases for both distributions (i.e., success rate of 0.72 and 0.76 respectively). On the 10-input size setup, all modules fail for all runs (i.e. have a success rate of 0) for both ranges.
***
**Experiment: More challenging distributions around 0**

Suggested by reviewer(s)/ relevant for reviewer(s):  **mB6g, ZiPP**

**Summary:** By training on various ranges using a truncated normal distribution, we found further failure cases for the modules (especially the RealNPU (modified)).

We trained using the following truncated normal distributions:
-	D1: train range= [-5,10] with mean=-1, stdev=3, test range= [-15,-5] with mean= -10, stdev=3
-	D2: train range= [-10,5] with mean=1, stdev=3, test range= [5,15] with mean= 10, stdev=3
-	D3: train range= [-5,5] with mean=0, stdev=1, test range= [5,15] with mean= 10, stdev=1

When trained using the 2-input setup, both NRU and SignNMRU have full success. The RealNPU (modified) has failure cases for all 3 distributions i.e., the success rates are D1=0.48, D2=0.64, D3=0.6.
When trained using the 10-input setup both the NRU and RealNPU (modified) have no success in any range. The NMRU does have success, but the success rate greatly varies depending on the range i.e., D1=0.48, D2=0.92 and D3=0.04.
***
**Experiment: RealNPU with L2 regularisation**

Suggested by reviewer(s)/ relevant for reviewer(s): **mhZR**

**Summary:** In general, using a L2 regulariser will perform worse than using L1.

Of the 9 tested ranges, L2 has a lower success rate than L1 for 5 ranges and has the same success rate for the remaining 4 ranges. If L2 regularisation is used instead of no regularization, it performs worse in 3 (of the 9) ranges, better on 3 ranges and the same on the remaining 3 ranges.

---

### Decision · Program_Chairs · 2021-09-27

**Decision:**

Reject

**Comment:**

This paper study possible adaptations of Neural Arithmetic Linear Units (Trask 2018) to better handle division.
While the contribution is worthwhile, the reviewers concern that it is too marginal and of limited scope